

# The atmosphere-land/ice-ocean system in the region near the 79N Glacier in Northeast Greenland: Synthesis and key findings from GROCE

Torsten Kanzow[1,2], Angelika Humbert[1,10], Thomas Mölg[3], Mirko Scheinert[4], Matthias Braun[3], Hans Burchard[5], Francesca Doglioni[1], Philipp Hochreuther[3], Martin Horwath[4], Oliver Huhn[2], Jürgen Kusche[6], Erik Loebel[4], Katrina Lutz[3], Ben Marzeion[7,8], Rebecca McPherson[1], Mahdi Mohammadi-Aragh[1,5], Marco Möller[8,12], Carolyne Pickler[3], Markus Reinert[5], Monika Rhein[2,7], Martin Rückamp[1,12], Janin Schaffer[1,13], Muhammad Shafeeque[7,8], Sophie Stolzenberger[6], Ralph Timmermann[1], Jenny Turton[3,9], Claudia Wekerle[1], and Ole Zeising[1]

[1]Alfred-Wegener-Institut Helmholtz-Zentrum für Polar- und Meeresforschung, Bremerhaven, 27570, Germany
[2]Institute of Environmental Physics, Department 1 of Physics and Electrical Engineering, University of Bremen, Bremen, 28334, Germany
[3]Institute of Geography, Friedrich-Alexander-University (FAU) Erlangen-Nürnberg, Erlangen, 91058, Germany
[4]TUD Dresden University of Technology, Chair of Geodetic Earth System Research, Dresden, 01062, Germany
[5]Leibniz Institute for Baltic Sea Research Warnemünde, Rostock, 18119, Germany
[6]University of Bonn, Institute for Geodesy and Geoinformation, Bonn, 53115, Germany
[7]MARUM, University of Bremen, Bremen, 28359, Germany
[8]Institute of Geography, University of Bremen, Bremen, 28359, Germany
[9]now at: Arctic Frontiers, Tromsø, 9007, Norway
[10]Department of Geoscience, University of Bremen, Bremen, 28359, Germany
[12]now at: Bavarian Academy of Sciences and Humanities, Geodesy and Glaciology, Munich, 80539, Germany
[13]Potsdam-Institut für Klimafolgenforschung (PIK), 14412 Potsdam, Germany

**Correspondence:** Torsten Kanzow (Torsten.Kanzow@awi.de)

**Abstract.**

The Greenland Ice Sheet has steadily lost mass over the past decades, presently representing the second-largest single contributor to global sea-level rise. Even the glaciers draining the Northeast Greenland ice stream have been observed to retreat and thin. Here, we present a comprehensive study of processes affecting and being affected by the mass balance of marine terminating and peripheral glaciers in Northeast Greenland. Our focus is on the 79N Glacier (79NG), which hosts Greenland's largest floating ice tongue. We provide new insight into the ice surface melt, ice mass balance, glacier dynamics, regional solid earth response, ocean-driven basal melt and the consequences of meltwater discharge into the ocean. Our study is based on observations, remote sensing and simulations with numerical models of different complexity, most of them originating from the Greenland Ice Sheet–Ocean Interaction Experiment (GROCE). We find the overall negative climatic mass balance of the 79NG to co-vary with summertime volumes of supraglacial lakes, and show the spatial pattern of overall negative ice mass balance for NE Greenland to be mirrored by the pattern of glacial isostatic adjustment. We find near coastal mass losses of both marine terminating and peripheral glaciers in NE Greenland to be of similar magnitude in the last decade. In contrast to the neighboring Zachariae Isstrøm, the 79NG – despite experiencing massive thinning of the floating tongue – has resisted





an acceleration of ice discharge across the grounding line due to buttressing imposed by lateral friction of the 70 km-long

ice tongue in the narrow glacial fjord. Observations and models employed in this study are consistent in terms of melt rates

occurring below the floating ice tongue. Our results suggest the multidecadal warming of Atlantic Intermediate Water flowing

into the cavity below the ice tongue – supplied by the recirculating branch of the West Spitsbergen Current in Fram Strait – to

be the main driver of the recent major increase in basal melt rates. We find the melt water leaving the cavity toward the ocean

at subsurface levels to quickly dilute on the wide shelf. The study concludes by summarizing important estimates of changes

to the state of the atmosphere, ice, land and ocean domains.

## 1  Introduction

About 25% of the current global mean sea level rise is caused by the mass loss of the Greenland Ice Sheet (Milne et al., 2009;

Groh et al., 2014; Rietbroek et al., 2016; Horwath et al., 2022), with a substantial increase of the mass loss rate in recent

decades (Shepherd et al., 2012; Khan et al., 2015; WCR, 2018; Löcher and Kusche, 2021) and particularly in very recent years

(Mankoff et al., 2021). The resulting rising Greenland freshwater input to the ocean could have significant implications for the

North Atlantic thermohaline circulation (Fichefet et al., 2003; Brunnabend et al., 2015; Martin et al., 2022). Decadal circulation

changes do not only affect regional sea level in the North Atlantic (Chafik et al., 2019), but also alter ocean temperatures which

in turn affect the marine ecosystems (e.g. through Herring recruitment; Toresen et al. (2019)).

What do we know about the reasons for the accelerating mass loss of the Greenland Ice Sheet? In their Special Report on the

Ocean and Cryosphere in a Changing Climate (SROCC), the Intergovernmental Panel on Climate Change (IPCC) emphasizes

that the increasing contribution of the Greenland Ice Sheet to sea level rise since the 1990s is related to the warming of the

surrounding oceans and atmosphere (Meredith et al., 2019). In fact, both the negative trend in surface mass balance and the

increased ice discharge into the ocean have contributed to the acceleration of mass loss after 2000 (Pattyn et al., 2018). The

strongest increase in mass loss has been observed at the margins and lower elevations of the Greenland Ice Sheet where

glaciers are retreating – accompanied by strongly accelerating glacier flow velocity and thinning (Howat et al., 2008; Rignot

and Kanagaratnam, 2006; Moon et al., 2012; Joughin et al., 2014; Helm et al., 2014). The contribution of peripheral glaciers

(i.e., glaciers dynamically disconnected from the inland ice sheet) to the total ice-mass loss of Greenland is large (about 25%

during 2000-2011; Bolch et al. (2013); Marzeion et al. (2015); and even more than 30% during 2003 to 2016, Horwath et al.

(2022)) despite the fact that they cover only 5% of the total ice area. This underlines the importance of processes between ice,

ocean, and atmosphere along the continental margins.

Observations of individual glaciers show that their acceleration and thinning during the recent past is related to a warming

of Greenland's fjord waters (Murray et al., 2010; Straneo et al., 2010, 2011; Mayer et al., 2018), caused by a subsurface inflow

of warmer ocean waters from the subtropical Atlantic onto the Greenland continental shelf (as shown in Fig. 1) – hereafter

referred to as Atlantic Intermediate Water (AIW) – below the cold and fresh Polar Water. Warmer water at the grounding line

and the calving front of marine terminating glaciers leads to an increased basal and frontal melt. This may cause the glacier

to thin and retreat, resulting in an acceleration of the ice flow upstream. Model studies agree in suggesting the ocean as a very



plausible driver of glacier changes (Vieli and Nick, 2011), but the underlying mechanisms are not yet fully understood and are thus not always realistically represented in geophysical models (see Straneo and Cenedese (2015), for a review). Spectacular evidence for ocean changes affecting Greenland glaciers can be found at Jakobshavn Isbræ – Greenland's largest glacier in

terms of ice volume discharge. After a 20-year phase of continuous, pronounced thinning and retreat, this glacier had started to strongly re-advance after 2016, supposedly as a consequence of anomalously cold water advected into the local fjord system (Khazendar et al., 2019). Some of the increasing submarine glacier melt over the past decades may, however, actually have been driven by increased subglacial runoff of meltwater from the ice sheet into the glacial fjords, particularly in West Greenland (Slater and Straneo, 2022).

What is the importance of mass loss by surface melt relative to that by increased ice discharge? Mouginot et al. (2019) found that more than 60% of the Greenland Ice Sheet mass loss has been due to ice discharge over the past 50 years. However, in recent years, the surface mass balance has played an increasingly important role, adding the atmosphere as a critical player (Meredith et al., 2019), interlinked by complex processes with the glacier surface (Mölg and Kaser, 2011). A temporary acceleration of ice mass loss in 2012 was linked to atmosphere-induced surface melt in southwest Greenland (Bevis et al., 2019), and air

temperatures on the central Greenland ice sheet in the first decade of this century appear to have been the highest throughout the past millennium (Hörhold et al., 2023). For floating ice tongue glaciers, the varying subsurface discharge of glacial meltwater forms another source of variability that partly originates from surface melt and enters the ocean at the grounding line and at the sloping ice base downstream, modulating the melt processes in the cavity beneath the floating ice tongue (Rignot et al. (2010); Motyka et al. (2011); Enderlin and Howat (2013); see schematic in Fig. 2).

So far, ice mass loss has been observed in all regions around the Greenland Ice Sheet margin, with strong contributions from marine terminating outlet glaciers (Rignot et al., 2010; Helm et al., 2014; The IMBIE team, 2018; Mouginot et al., 2019). The retreat of these glaciers coincides with the warming of AIW (e.g., Straneo et al. (2010)). In recent years the warming of the North Atlantic has also penetrated into the Nordic Seas and the Arctic Ocean (Beszczynska-Möller et al., 2012; McPherson et al., 2023), amounting to almost 1° C warming in the West Spitsbergen Current (WSC) between 1997 and 2018.

The collapse of the floating ice tongue of Zachariæ Isstrøm (hereafter ZI) in NE Greenland is one of the most prominent examples of rapid glacier retreat (Mouginot et al., 2015). Currently, the largest remaining floating ice tongue in Greenland is the 80 km-long one of the 79 North Glacier (hereafter 79NG) located just north of ZI. Studies based on remote-sensing data report melting and thinning of the 79NG glacier tongue (Kjeldsen et al., 2015; Mouginot et al., 2015), with estimates of up to 30% total ice thickness loss over the past 20 years (Mayer et al., 2018). This thinning appears to be caused by a mass imbalance

due to increased melt along the ice tongue base (Wilson et al., 2017; Mayer et al., 2018) caused by ocean heat fluxes. Based on water mass analyses, Schaffer et al. (2017) were able to demonstrate the existence of a pathway of warm AIW recirculation in Fram Strait across the NE Greenland continental shelf towards 79NG. They also showed that the AIW on the continental shelf close to 79NG has warmed by 0.5° C over the past few decades, driven by the advection of the warming Atlantic Water in the WSC across Fram Strait which penetrates onto the continental shelf (McPherson et al., 2023). In contrast, the reason for the

collapse of the floating ice tongue of ZI is less clear (Mouginot et al., 2015).




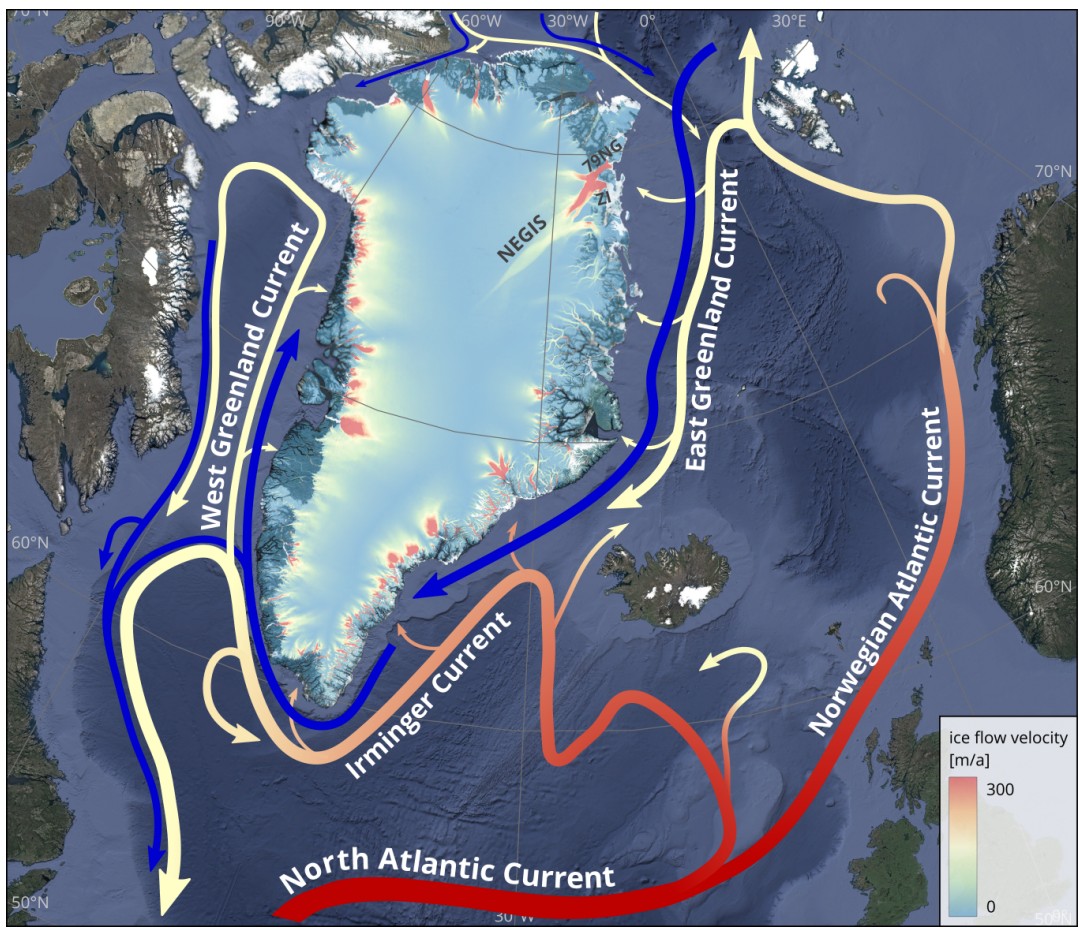

**Figure 1.** The Greenland Ice Sheet with ice-flow velocity (color shaded; Joughin (2015)), the topography of the surrounding ocean and land areas, and the main ocean currents. The red arrows denote warm and saline Atlantic Water which progressively cools along its flow path, depicted by the color changing from red (warm) to yellow (cooler). When this water subducts below the cold and fresh Polar surface water (the latter denoted by blue arrows) exported from the Arctic along the continental shelf break of eastern, southern and western Greenland (yellow arrows crossing underneath the blue ones), we refer to it as Atlantic Intermediate Water (AIW). The Northeast Greenland Ice Stream (NEGIS), which separates into 79NG, ZI and Storstrømmen at the ice sheet margin, is seen as a stream of increased ice-flow velocity in NE Greenland. For a zoom into the ice sheet–ocean system in NE Greenland see Fig. 3.

The need to assess and understand the changing ice sheet–ocean system of NE Greenland was the starting point of the GROCE project. GROCE was funded by the German Ministry of Education and Research (BMBF). It consists of scientists from several universities and research centers joining their expertise in observations, remote sensing and modeling of ocean and glacier dynamics as well as processes in the atmosphere and the lithosphere on a wide range of scales. The overall aim of
GROCE has been to understand critical processes and their interactions in controlling the NE Greenland Ice Stream (NEGIS) ocean–glacier interaction, with a focus on 79NG.





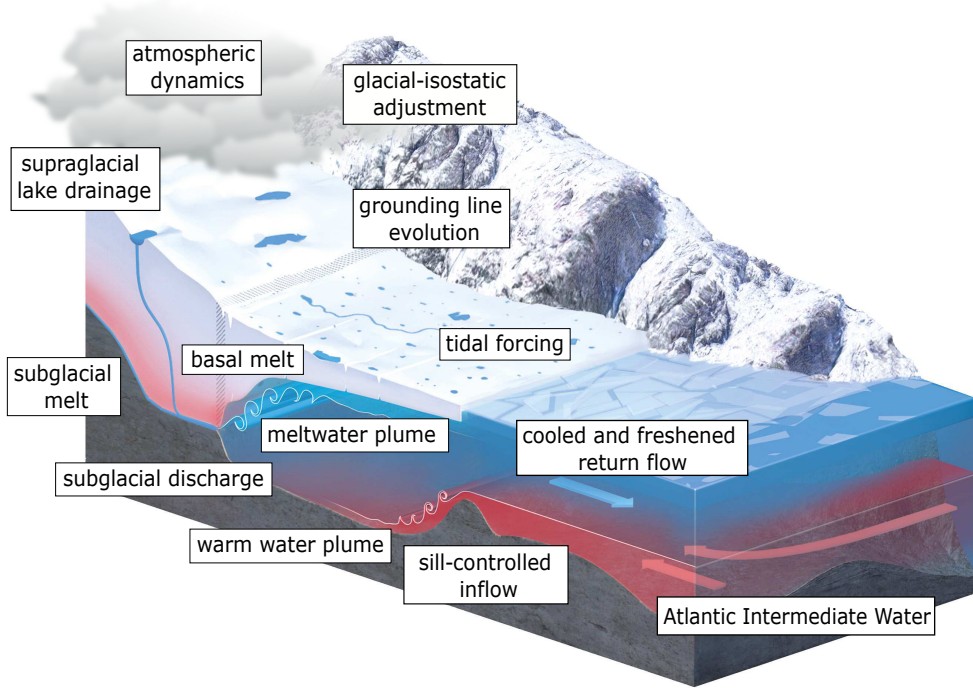

**Figure 2.** Graphic depiction of the NEGIS/79NG ice sheet-ocean system showing the ocean, atmosphere, land, and glacier processes involved in ocean–glacier interactions, inspired by our results regarding the mass balance of the 79NG. The floating ice tongue is a prominent characteristic of the 79NG, affecting glacier dynamics and ocean-driven glacier melt.

As pointed out above, there are a variety of studies that provide ice mass balance estimates for Greenland. The aim of this paper is to provide a detailed, process-based understanding of the ice mass balance of the NEGIS-79NG system and a synthesis of processes from atmosphere to ice, to ocean and to lithosphere. In Chapter 2 we will briefly introduce the GROCE project. Chapter 3 contains a summary of the data sets and methods used. Chapter 4 presents the results starting from the atmosphere, and moving on via cryosphere and land domains to finally consider the ocean processes. In Chapter 5 we will integrate the results into a consistent view of the NEGIS-79NG mass balance.

## 2 The GROCE project

Thematically connected studies of local processes and observational programmes in the ocean, on the 79NG and at adjacent land areas of NE Greenland form the basis of our research. Fig. 2 highlights the variety of processes relevant for the coupled system. The consortium covers atmospheric processes affecting the surface melt and the drainage of meltwater from supraglacial lakes to the glacier base. We elucidate the effects of the drained surface meltwater entering the cavity across the



grounding line (subglacial discharge) and (ii) of the warm AIW entering the cavity from the NE Greenland continental shelf on

the meltwater plume dynamics and basal melt in the cavity and on frontal ablation (Fig. 2). We further investigate the impact

of tides and ice-shelf retreat on the ice-sheet velocity field upstream, and we shed insight into the role of the distribution of

meltwater outflow on the regional ocean circulation. Furthermore, we investigate the impact of the glacial-isostatic adjustment

on the complex solid earth–ice–ocean system, particularly in the marginal zones of the NEGIS.

## 3   Data basis and methods

In this chapter, we briefly introduce the data and methods applied. Since data sets and methods have been previously published

in peer-reviewed papers to a large extent, we keep the description of each element short and refer to the literature for more

detailed information. Fig. 3 and 4 summarize the data sets and their geographical origins in the 79NG area.

### 3.1   New multi-year time series of climatic mass balance of the NEGIS inferred by regional modeling

The basis of inferring the climatic mass balance is the polar version (Hines et al., 2011) of the Weather and Research Forecasting

(WRF) model. WRF is a non-hydrostatic numerical atmospheric model that has been extensively used and tested at many scales

and in various regions (Powers et al., 2017). Our applications (version 3.9.1.1) concerned (1) a case study to examine processes

of wintertime melt over $\sim$20 days in 2014 (Turton et al., 2019) and (2) the generation of a new high-spatially (1 km) and

temporally (up to 1hr) resolved atmospheric modelling dataset over the 79NG and NE Greenland Ice Stream for 2014-2018,

called NEGIS_WRF (Turton et al., 2020).

The output from this simulation serves as forcing for the recently developed COupled Snowpack and Ice surface energy and

mass-balance model in PYthon (COSIPY) (Sauter et al., 2020). This model calculates the mass balance at the surface and near

the surface (water storage and melting/refreezing in the snow), the sum of which is known as "climatic" mass balance (CMB).

The COSIPY/NEGIS_WRF setup (Fig. 5) delivered multiannual fields of the CMB every 1 km for the 79°N region (Blau et al.,

2021), which in the GROCE framework represents the atmosphere-cryosphere interface.

### 3.2   Meteorological data

Hourly-resolution records from automated weather stations (AWS) were utilized to examine the atmospheric conditions in

the region of the 79NG and evaluate the modeling above (purple diamonds in Fig. 3). AWS 9602 and AWS 9604 from an

earlier field program provided data for the mid to late 1990s on the floating shelf (Højmark Thomsen et al., 1997), while two

AWS from the Programme for Monitoring of the Greenland Ice Sheet (PROMICE) network delivered data since the late 2000s

for Kronprins Christian Land (KPC). Near-surface air temperature, relative humidity, wind speed and direction, air pressure,

shortwave radiation, longwave radiation, cloud cover and skin temperature were among the variables considered from AWS.

The European Centre for Medium-Range Weather Forecasts (ECMWF) ERA-Interim reanalysis dataset (Dee et al., 2011)

was used alongside the AWS observations to compensate for their sparse and intermittent observing period. The reanalysis





**Figure 3.** Geographic overview over the near-coastal NEGIS (79NG and ZI) area out to the shelf break of NE Greenland. Also shown are the geographic locations of in-situ data and remote sensing datasets used in and produced by the GROCE project in the ocean on the continental shelf, on the floating ice tongue of the 79NG, on the grounded ice and on the solid earth. Grey arrows denote the pathway of the AIW circulation in the ocean along the Norske Trough-Westwind Trough system.

data span 1979 to 2017 and were evaluated against the AWS data to ensure they capture the climatological characteristics of the NEGIS/79NG region. Also, ERA-Interim provided lateral boundary conditions for the atmospheric modeling (Sect. 3.1).



**Figure 4.** Detailed map focusing on 79NG and ZI and the inner continental shelf (zoom into map shown in Fig. 3). The ice flow velocities are based on Landsat-8 imagery recorded in 2016 (processing according to Rosenau et al. (2015)). Calving fronts were inferred from Landsat-8/9 imagery (see Sect. 3.6). Annotated red rectangles denote the location of supraglacial lakes where in-situ measurements of water depths were taken.

## 3.3 New data set of evolving supraglacial lake volumes

To quantify the amount of meltwater gathering on the glacier surface, and subsequently draining to the glacier bed (see Fig. 2) or running off through surface channels into the ocean, we developed a method to identify the dynamic boundaries of supraglacial lakes (Hochreuther et al., 2021) which we apply to estimate lake areas. This method uses Sentinel-2 multispectral data to segment the lakes on a daily basis from 2016 to 2022, and a thresholding technique, taking advantage of the prominence of lakes in the blue and red spectral bands. We extend our previous research by using a widely accepted supraglacial lake depth algorithm to provide a rough volume estimate of the lakes. Originally proposed by Philpot (1989) and further developed by Sneed and Hamilton (2007), this method employs the principles of radiative transfer theory to determine the lake depth, $z$,





from an optical image by

$$z = (\ln(A_d - R_\infty) - \ln(R_w - R_\infty))/(-g), \tag{1}$$

where $A_d$ is the lake bottom albedo, $R_\infty$ is the reflectance for optically deep water, $R_w$ is the observed water reflectance, and $g$ is the two-way attenuation coefficient. The pixel-wise depth output from this equation is then multiplied by the area of the pixel to calculate the respective volume. Here, the green band is used as the basis of the calculations. $R_\infty$ is empirically determined to be 0.0866 from optically deep water present in several Sentinel-2 scenes, and the constant of $g$ is 0.1413 (Williamson et al., 2018). Furthermore, the value of $A_d$ is calculated for each lake by averaging the green reflectance values extending 90 meters radially around its perimeter. In order to reduce the noise in the lake volume time series, the estimates are exclusively based on cloud-free conditions, which are subsequently interpolated onto a regular time grid. Finally, for smoothing purposes, a Savitzky-Golay filter is applied to the lake volume time series. The methodology is applied to all supraglacial lakes contained in the area just upstream of the grounding line of 79NG area (pink shaded area in Fig. 3).

### 3.4 Updated method and new data to study the tidal response of glaciers

Observed tidal motion of the floating tongue of the 79NG (Fig. 4) is used as a forcing for the subglacial hydrology model CUAS (Beyer et al., 2018) and viscoelastic simulations using COMice-ve (Christmann et al. (2021); Fig. 5). The CUAS setup uses an updated version of the BedMachine topography (Morlighem et al., 2017) including AWI's flight lines in the 79NG region acquired within the GROCE project (Christmann et al., 2021) and parameters as used in Beyer et al. (2018). CUAS simulations are conducted at a resolution of 300 m in the vicinity of the grounding line of the 79NG and of 1200 m over the entire NEGIS. We consider water input by basal melt in the grounded area arising from geothermal flux and basal friction taken from Aschwanden et al. (2016). The tidal forcing (Fig. 2) is expected to modulate the effective pressure in the CUAS simulation (mostly at the grounding line) which is fed into COMice-ve to estimate the consequences on the ice flow.

### 3.5 Determination of frontal positions and ice-flow velocity fields by an advanced deep neural network

The processing comprises data of the remote sensing satellites Landsat-8 and Landsat-9. The inference of ice-flow velocity fields is based on the feature tracking method as implemented and described by Rosenau et al. (2015). For the inference of frontal positions a deep neural network was implemented (Loebel et al., 2022). A thorough assessment was carried out supplementing the single-band inputs with multispectral data, topography, and textural information. The resulting feature importance shows a clear benefit utilizing multispectral bands. With their inclusion, frontal position predictions are generally more accurate than using conventional single-band inputs only (Loebel et al. (2022); see Fig. 4). As a result, we could infer about 150 flow-velocity fields per year for 79NG and ZI as well as about 40 to 60 frontal positions per year for each glacier. Additionally, in order to assess the change of the frontal position in a better way a box method was implemented which enables to track the proportionate (glacier-covered) area as time series where decreasing area equals a retreat of the glacier front (Loebel et al., 2023).




## 3.6 New and more detailed sensitivity studies of the 79NG ice discharge

To assess the upper-end effect of the ice discharge of the 79NG on sea level change we conduct a series of model simulations of three scenarios by a subsequent removal of increasingly larger frontal parts of the 79NG floating tongue up to a full collapse (i.e. the removal of the entire tongue), following Humbert et al. (2023). The simulations using the ice sheet model ISSM (Larour et al., 2012) and the initial state are based on a joint inversion for the basal friction coefficient on grounded ice and the ice hardness in floating ice. The friction law is of non-linear Budd type (Budd et al., 1984). The geometry is based on BedMachine

v4 (Morlighem et al., 2017). We define three future states: (1) calv-iceberg: assuming a calving of a larger part at the eastern ice front, (2) calv-fjord: a retreat up to a bottleneck within the fjord leaving a 45 km long tongue, and (3) a total collapse of the floating tongue. We analyse the 3D simulations as cross-sections in along-flow direction and across the grounding line. The variables discussed here are surface velocity, ice flux across the grounding line and buttressing.

## 3.7 A refined approach to study the mass balance of the Greenland Ice Sheet

We combined two major techniques for observing ice sheet mass balance (Kappelsberger et al., 2021). Firstly, we analysed results from GRACE and GRACE-FO satellite gravimetry, which is directly sensitive to mass changes, using the monthly solutions inferred by Bettadpur (2018) and by Kvas et al. (2019) given in a spherical harmonic representation. We applied the method of tailored sensitivity kernels (Groh and Horwath, 2021) which integrates the concepts of region definition, filtering according to known noise characteristics, and minimisation of leakage effects. The effect of glacial-isostatic adjustment (GIA)

is corrected for by subtracting the mass effect predicted by the respective GIA model (see Subsections 4.7 and 4.8). Secondly, satellite altimetry measures ice-surface elevation and allows to infer surface-elevation change (SEC). We used CryoSat-2 data pre-processed with regard to the acquisition mode (LRM or SARIn) according to Krieger et al. (2020) and Helm et al. (2014), respectively. We conducted a repeat altimetry analysis (Kappelsberger et al., 2021) over the GrIS and the peripheral Flade Isblink. This analysis provides SEC rates in a 1.5x1.5 km$^2$ grid, determined by least-squares adjustment together with further

parameters accounting for local topography, seasonal variation, and the time-variable radar penetration effect. For the peripheral glaciers except Flade Isblink Ice Cap (FIIC), the SEC product published by Simonsen and Sørensen (2017) was used. The effect of GIA bedrock uplift was corrected for taking predictions by respective GIA models into account (see Subsections 4.7 and 4.8). Volume changes were converted to mass changes by employing a density model based on both the firn densification model IMAU-FDM (Ligtenberg et al., 2018) and on a velocity-based criterion for assuming dynamic ice mass imbalances. Finally,

the results (including uncertainty ranges) from the two complementary techniques were combined in a consistent parameter estimation approach (Kappelsberger et al., 2021). The resulting new grid of mass-change rates (given over the period 2010 to 2017) maintains the high spatial resolution of satellite altimetry and, at the same time, adheres to the mass change constraints provided by satellite gravimetry.



## 3.8 Response of the solid Earth to mass changes

The bedrock displacement (Fig. 2) was determined by in-situ measurements using geodetic GNSS recordings in the ice-free regions of NE Greenland. We carried out campaign-style measurements at ten sites between 78° N and 81° N in 2008/2009, 2016 and 2017 (Fig. 3). Additionally, data of the continuously recording GNET stations (Khan et al., 2016) were included. The GNSS data were analysed using the differential processing approach incorporating the required precise data and correction models (Kappelsberger et al., 2021). The solution refers to the GNSS-only realization IGS14 (Rebischung et al., 2016) of the International Terrestrial Reference System, with CM (center of mass of the Earth system) adopted as origin. Finally, 3D linear velocities were inferred with uncertainties of 1 mm a$^{-1}$ for the horizontal components and 1.5 mm a$^{-1}$ for the vertical component (Sect. 4.8).

## 3.9 A refined approach to model peripheral glaciers

We used the Open Global Glacier Model (OGGM; Fig. 5) (Maussion et al., 2019) to simulate Flade Isblink Ice Cap (FIIC) in the vicinity of the 79NG in NE (NE) Greenland (Fig. 3). The Randolph Glacier Inventory (RGI) represents FIIC as a single glacier entity, classified as ice cap and as marine-terminating (Consortium, 2017). We subdivided FIIC based on the ArcticDEM data (Porter et al., 2018) into 299 individual glacier basins (Fig. 12a). Using different ice velocity data (Gardner et al., 2022; Joughin, 2015), six out of nine marine-terminating basins were detected as active calving basins (Recinos et al., 2021). The frontal ablation of marine-terminating glaciers was computed using a dynamic parameterization based on the calving law formulated by (Oerlemans and Nick, 2005) and applied at large scale by (Huss and Hock, 2015). However, we refined this approach by incorporating mass conservation principles to estimate the thickness of the calving front, which was determined by reconciling the amount of ice delivered to the terminus by OGGM with the amount of ice calved (Recinos et al., 2019).

We also used the new NEGIS_WRF atmospheric data (Sect. 3.1) from 2014 to 2018 at 5 km resolution for one of the OGGM simulations. Additionally, the datasets ERA5 (~30 km) and CRU (~50 km) (Harris et al., 2020) were employed in order to analyze the sensitivity of glacier processes to input data resolution. Similarly, gridded rates of ice-mass change based on the combination of satellite gravimetry (GRACE) and altimetry (CryoSat-2) at a fine resolution of 1.5 km × 1.5 km (Kappelsberger et al., 2021) (Sect. 3.7) were used to calibrate OGGM for the new glacier basins outlines of FIIC (Fig. 12c). Geodetic mass balance data (Hugonnet et al., 2021) were used for the rest of the peripheral glaciers in NE Greenland. Finally, future glacier mass loss and freshwater runoff contributions were projected for all peripheral glaciers in NE Greenland using CMIP6 (10 GCMs and 4 emission scenarios) climate forcing and OGGM. The selected GCMs have been employed in several previous studies (Rounce et al., 2023; Zhao et al., 2023; Edwards et al., 2021; Malles et al., 2023). A standardized selection of GCMs provides consistency and continuity in the research and facilitates the comparison and contrast of results (Marzeion et al., 2020; Hock et al., 2019). Furthermore, the selected GCMs have a large enough sample size to encompass a wide range of potential climatic futures, thus yielding a robust set of scenarios and increasing confidence in the projections. As in the studies mentioned above, the selected GCMs were bias corrected using the delta method (Maraun, 2016; Trzaska and Schnarr, 2014), which involved employing relatively high-resolution gridded observations as reference



climatology (Copernicus Climate Change Service, 2019), and applying only anomalies from the GCMs relative to a pre-determined reference period (1981-2019). Ice discharge from Basins 2 and 3 of FIIC (Fig. 12b) forms a continuous ice shelf that extends towards the northwest. We determined the grounding line locations of these basins and analyzed their variability
with time (Möller et al., 2022).

## 3.10    Observations of Atlantic Intermediate Water flow toward the 79NG and inferred basal melt rates

Since 2014, dedicated oceanographic measurements have been carried out on the continental shelf of NE Greenland relevant for the ocean impact on the 79NG. During the Polarstern expedition PS85 in 2014, an array of seven moorings was deployed across Norske Trough (Fig. 3) – the inflow pathway of warm AIW toward the 79NG (Schaffer et al., 2017). The array was
recovered in summer 2016 during the expedition PS100. Two two-years-long time series of temperature, salinity and water velocity lay the foundation of the study of AIW circulation and connection from the shelf edge to mid-shelf (Münchow et al., 2020). The expedition PS100 was also used to deploy moorings (measuring velocity, temperature and salinity) in the direct vicinity of the 79NG for the first time. The mooring deployment had been guided by hydrographic and bathymetric surveys close to the calving front of the 79NG (Schaffer et al., 2020), which revealed a complex system of sills and channels guiding
the flow of AIW into the cavity under the floating ice tongue of the 79NG (Fig. 4). The moorings were recovered and partly re-deployed in 2017 during the Polarstern expedition PS109. The most recent recovery of the moorings close to the 79NG took place in summer 2021 aboard the Danish coastguard vessel HDMS Triton. The data were used to study pathways, volume and heat transport and underlying dynamics associated with the AIW flow toward the 79NG including estimates of basal melt rates (Schaffer et al., 2020; von Albedyll et al., 2021). The bathymetry data mainly collected during the expedition PS100 was used
to update a global bathymetry grid (RTopo-2.0.4; Schaffer et al. (2019)), which then fed back into the FESOM ocean-sea ice model configuration developed in the project (Sect. 3.13).

## 3.11    Basal melt of 79NG's floating tongue from in situ radar observations

Basal melt rates are measured using an autonomous phase sensitive radar (ApRES) instrument that acquired data from July 2018 to July 2022 continuously. From July 2018 to July 2020 the instrument moved with the glacier over a distance of about 4.5
km, making this a Lagrangian observation (magenta line in Fig. 4). In July 2020 the instrument was relocated to the original start location July 2018, such that the trajectory was occupied twice, so that temporal changes in basal melt might be distinguishable from spatial ones.The instrument was deployed outside the hinge zone, about 7.7 km downstream from the grounding line, but only 3.6-4.7 km away from the southern margin of the floating tongue towards Lambert Land. We basically observe the change in ice thickness over time by tracking the basal reflection and assessing the influence of stretching by deformation. More details
on the method are given in Zeising et al. (2023). As the base of the floating tongue is rough, side reflections are influencing our ability to detect the basal melt rate in nadir direction.



### 3.12  Quantification of the basal meltwater from noble gas data

To identify and to quantify the distribution of basal meltwater on the NE Greenland shelf, we use helium (He) and neon (Ne) observations. They have been sampled during the expeditions PS100 (in 2016) and PS109 (in 2017) on the NE Greenland Shelf. On the R/V Maria S. Merian expedition MSM85 (2019), hydrographic and tracer sections across the East Greenland Current from the Irminger Sea to 79NG were taken to put the results from the 79NG in a broader perspective (Mertens et al., 2020). The calculation of the basal meltwater follows the approach described in detail in Rhein et al. (2018) and Huhn et al. (2021). Atmospheric air with a constant composition of the noble gases He and Ne is trapped in the ice matrix during formation of the meteoric ice. When the ice melts at depth or at the ice base inside a glacier cavity, these gases are completely dissolved in the water, due to the enhanced hydrostatic pressure. This leads to an excess of DHe = 1280% and DNe = 890% in pure basal melt water (Loose and Jenkins, 2014); the D stands for the gas excess over the air-water solubility equilibrium. Instead, melt at the glacier surface would equilibrate quickly with the atmosphere and does not show a noble gas excess in ocean water.

Basal meltwater may be additionally enriched in crustal He (i.e., the isotope 4He) from $\alpha$-decay of uranium, thorium, and their daughter products in the bedrock beneath an ice sheet. There, He can be enriched 3 to 4.5 times compared to pure basal meltwater (Craig and Scarsi, 1997; Jean-Baptiste et al., 2001). 4He enriched meltwater with a significantly enhanced He/Ne ratio may originate from upstream the grounding line, i.e. from glacial ice that accumulates 4He and melts by friction or geothermal heat and subsequently enters the cavity as subglacial discharge. Alternatively it may originate from melt downstream but near the grounding line, where the deepest and oldest ice is located. Instead freshwater discharged into the cavity originating from supraglacial melt does not carry a noble gas signal and is thus not detectable by this method (Huhn et al., 2021).

### 3.13  Multi-scale approach to comprehensively simulate cavity circulation, basal melt, and spreading of basal meltwater

We take a multiscale approach in simulating the basal melt associated with 79NG (Fig. 5). Idealized (i) 2D and (ii) 3D approaches are taken to understand the governing processes in the glacier cavity driving the basal melt (Fig. 2) and influencing the melt rate distribution. (iii) A high-resolution configuration of the Finite-volumE Sea ice–Ocean Model (FESOM2.1) with a realistic representation of the cavity geometry of the 79NG is used to study changes of the melt rates in response to a hierarchy of driving factors (most importantly, AIW temperature on the NE Greenland continental shelf and subglacial discharge into the cavity). (iv) Another FESOM setup with increased mesh resolution around Greenland is then used to study the impact of glacial meltwater from the ice sheet on circulation and hydrography in the Nordic Seas and the Subpolar North Atlantic. In the following the different models are introduced.

**2D fjord model:** Classical ocean models have inherent problems in resolving subglacial meltwater plumes (Fig. 2), which are buoyant turbulent gravity currents underneath ice shelves modified by subglacial discharge, basal meltwater, and the entrainment of warm AIW. Typical meltwater plume thicknesses are on the order of 10 m, and a plume can cover water depths reaching from the grounding line at several hundreds of meters depth to the sea surface. A recent one-dimensional modeling





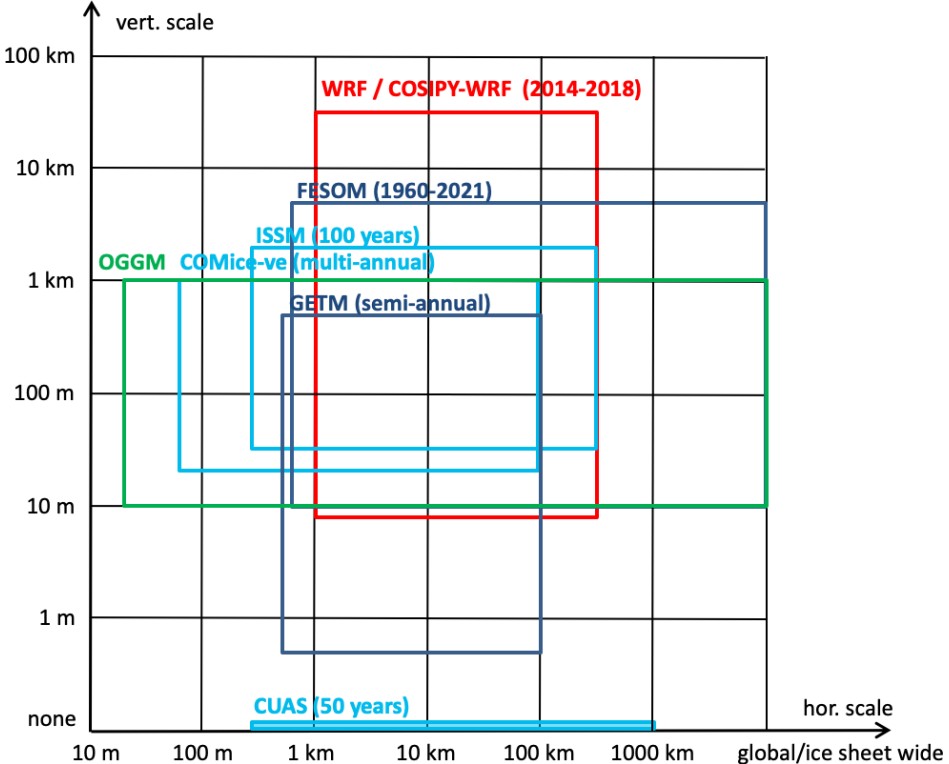

**Figure 5.** Horizontal and vertical ranges of the ocean, glacier, ice sheet and atmospheric models used in this study. For each model (rectangles), the lower left corner denotes the horizontal and vertical resolutions (model grid size) while the upper right corner denotes the horizontal and vertical extents of the model domains.

295    study (Burchard et al., 2022) showed that the vertical model resolution inside the meltwater plume and the entrainment layer at its base should be 2 m or better. For geopotential-coordinate models (e.g., Losch (2008)), this would have the consequence of an overall vertical model resolution of at least 2 m, which is infeasible for global models. A solution to this problem are models with vertical coordinates that follow the ice–ocean interface, such as terrain-following sigma-coordinate models (e.g. Beckmann et al. (1999); Dinniman et al. (2007); Timmermann et al. (2012)). While the use of sigma-coordinates under ice shelves

300    generally solves the vertical resolution problem at the ice–ocean interface, the resolution inside the meltwater plume would be proportional to the thickness of the water column from the ice–ocean interface to the bottom. To reduce this dependency, we use adaptive vertical coordinates (Hofmeister et al., 2010). We constructed an idealized two-dimensional model of the 79NG fjord with high resolution in the vertical direction and along the fjord axis (Reinert et al., 2023), using the coastal ocean model GETM (Burchard and Bolding (2022); Fig. 5). The vertical coordinates adapt to the stratification in a way that coordinate layers

305    accumulate at locations of enhanced vertical stratification (see for applications in Hofmeister et al. (2010, 2011); Gräwe et al. (2015); Li et al. (2022)). This strongly reduces several problems of sigma-coordinates, for example the dependence of the layer thickness on the water depth as well as the pressure-gradient error (Hofmeister et al., 2010).





**3D fjord model:** As a next step towards a realistic three-dimensional simulation of circulation and basal melt under the floating ice tongue of 79NG, we added to the 2D fjord model a third dimension, the across-fjord axis. This yields an idealized 3D model of the 79NG fjord with a high resolution of 500 m in both horizontal directions and adaptive coordinates in the vertical.

**2D plume model:** To study the impact of small-scale features at the ice base on basal melt, Mohammadi-Aragh and Burchard (2024) developed a 2D plume model, the General Ice shelf water Plume Model (GIPM). In contrast to the 2D fjord model, here the two dimensions span the base of 79NG. For this study, an equidistant grid resolves the domain of interest with a 150 m-resolution. We use a realistic ice-base topography of 79NG, guided by the BedMachine Greenland v5 dataset (Morlighem et al., 2017). The study encompassed two experiments exploring the impact of small-scale feature in ice base topography on the basal melt rate, one with a relatively smooth and the other with a more realistic (i.e. rougher) ice base, featuring elongated subglacial channels inspired by observations (Zeising et al., 2023). The potential temperature and salinity of the ambient water (here AIW) as ocean forcing are taken from observations obtained during the expedition PS100 close to the calving front of the 79NG (Kanzow, 2017).

**3D ocean–sea ice model:** For simulations of the ocean circulation in the cavities of 79NG and ZI and of the spreading of glacial meltwater around Greenland, we used two global setups of the Finite-volumE Sea ice-Ocean Model (FESOM2.1) and the Finite Element Sea ice–Ocean Model (FESOM1.4), respectively (Fig. 5). An advantage of the FESOM model family is its multi-resolution capability in a global framework, which allows us to easily adapt the mesh resolution to the area of interest. FESOM2.1 includes an ice-shelf component explicitly resolving the 79NG and ZI cavities (Wekerle et al., 2024). The thermodynamic interaction between ocean and ice shelf (i.e. ice shelf basal melt) is simulated based on the three-equation system which has been proposed by Hellmer and Olbers (1989) and implemented in FESOM by Timmermann et al. (2012). It computes salinity and temperature at the ice shelf–ocean boundary layer from diagnostic heat and salt budgets. The horizontal resolution of the mesh in the vicinity of the 79NG and ZI, the NE Greenland continental shelf and in the Arctic Ocean was set to 0.7 km, 2.5 km and 4 km, respectively. For vertical discretization, 86 geopotential levels were used, with 5 m layer thickness in the top 100 m and coarser layer thickness towards the seafloor. Bathymetry and ice shelf topography were taken from the RTopo-2.0.4 data set (Schaffer et al., 2019). In the model, freshwater runoff from the Greenland Ice Sheet was injected into the ocean along the coast and along the grounding lines of the 79NG and ZI cavities (i.e. as subglacial discharge within the cavity). Thereby, solid ice discharge was derived from (Mankoff et al., 2020b) and liquid water discharge from (Mankoff et al., 2020a). Both data sets have daily temporal and high spatial resolution (∼100 m). A simulation has been conducted for the years 1960-2021, forced by the JRA55-do-v1.4.0 atmospheric reanalysis data set. The simulation shows good agreement with observed AIW temperatures and circulation in Norske Trough (McPherson et al., 2023), inflow of Atlantic Intermediate into the cavity and basal melt rates.

The spreading of glacial meltwater in the oceans around Greenland has been simulated with FESOM1.4 (Stolzenberger et al., 2022). The global mesh used in this study has a broader focus on the area around Greenland, the Arctic Ocean, Nordic Seas and northern North Atlantic. Mesh resolution in this area was set to 6 km, which can be considered as 'eddy-permitting'. The cavities of the 79NG and ZI are not included in this configuration. FESOM1.4 was initialized with temperature and salinity





climatologies provided by the Polar Science Center and forced with atmospheric fields of the JRA-55 reanalysis (Kobayashi et al., 2015). As we were interested in the effects of Greenland freshwater on the surrounding ocean, we performed two
experiments: i) Greenland freshwater is discharged into the ocean according to Bamber et al. (2018) and ii) the Greenland freshwater discharge is set to zero. With these experiments, we are able to analyze the meltwater signatures in the ocean. The two simulations cover the years 1980 to 2016, with 1980–1993 being the spin-up phase.

## 4    Main findings at the process-level

The aim of this paper is to provide a process-based understanding of the mass balance of the NEGIS-79NG system. We will
proceed to present the main results of GROCE in this regard, starting with atmospheric and ice surface processes, then moving on to glacier and solid earth aspects and finally consider ocean impacts.

### 4.1    Atmospheric processes and effects on the glacier surface

An investigation into the atmospheric processes in the region of the 79NG shed light on their role for surface melt. Using a combination of the data in Sect. 3.1, a clear surface-air temperature ($T_a$) increase of 3˚ C over 1979-2017 was identified (Turton
et al., 2019). This tendency agrees with, yet exceeds the global trend. Interestingly, significant winter temperature variability is noted with high-amplitude short-lived events ($T_a > 10$˚C over 48h) occurring each year between November to March over the study period. These have been linked to two mechanisms: 1) warm-air advection and 2) mixing from katabatic winds. Over 15% of these warm-air events were found to occur in March and could result in a short period of ice melt as $T_a$ increases along with solar radiation. On the other hand, summer $T_a$ varies little, which has been attributed to high pressure over Greenland and
katabatic winds dominating the wind direction. Turton et al. (2021) examined summer temperature variations and their link to supraglacial lake or surface melt ponds in the region for 2016-2019. The above-average air temperatures of 2016 and 2019, along with a number of rain events, led to extensive supraglacial lake formation up to elevations of 1600 m. Colder summers with a large accumulated snowpack, for instance in 2018, led to a limited formation of supraglacial lakes, limited to elevations of 800 m. Furthermore, a comparison of these findings with the total area of lakes from the early 2000s (Sundal et al., 2009)
points to an inland expansion and increase in area. More characteristics of the supraglacial lakes are shown in Sect. 4.3.

### 4.2    Atmosphere-driven changes in the climatic mass balance

By applying the new high-resolution NEGIS_WRF atmospheric data to the COSIPY model (Sect. 3.1) for 2014-2018 at 1 km resolution, the simulations revealed strong interannual variability in the CMB (Blau et al., 2021). The 4-year mean specific CMB of the 79NG in the defined region (here, inland to ∼30° W; light blue shading in Fig. 3) amounts to a mass loss of 744
mm a$^{-1}$, which is driven by runoff of melt water produced at the surface or inside the snow layers (evaporation is negligibly small). However, as shown in Fig. 6, the annual CMB range showed mass losses from as much as 1049 mm (2016/17) down to 120 mm (2017/18). The higher, almost balanced CMB in 2017/18 was a result of the cooler summer conditions and coincided with rather weak supraglacial lake activity (see Sect. 4.1). The strong 2016/17 mass loss, in turn, was associated with strong





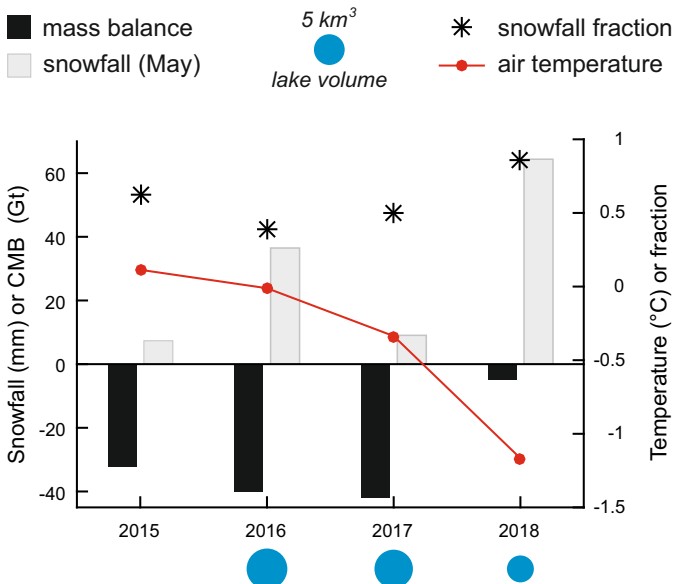

**Figure 6.** Area-wide climatic mass balance (October of previous year to September of current year), area-averaged 2-m air temperature and snowfall fraction (June-July mean) and snowfall amount (May), and daily max. lake volume (available since 2016, see Sect. 4.3) of 79NG area for the period 2014/15 to 2018 (when data sets overlap). Note that in the main text, CMB is given in mm a$^{-1}$ rather than Gt a$^{-1}$

.

surface melt that enhanced the ablation and runoff. The difference in surface albedo was found to be a major factor between
contrasting CMB years (Blau et al., 2021); in 2018 for example, high albedo was driven by higher accumulation during summer
(including the early ablation season) due to the lower air temperatures, which resulted in more frequent snowfall and a higher
snowfall fraction. Hence, summer precipitation amounts alone does not explain CMB variability sufficiently. Fig. 6 summarizes
the above mentioned relations.

### 4.3 Supraglacial lake volume evolution

Supraglacial lakes are one manifestation at the atmosphere/glacier interface, as mentioned in Sect. 4.1. Here, the combined
effect of air temperature, snowpack thickness and rainfall on the development rate of supraglacial lakes is directly observed.
Melt seasons with a combination of higher air temperatures, thinner snowpack or more rainfall lead to larger average lake areas
and a more widespread distribution of lakes at higher altitudes. In Hochreuther et al. (2021), the total lake area upstream of the
79NG and ZI grounding lines (displayed in Fig. 3 as pink shading) is extracted and tracked over each of the investigated melt
seasons. The method described in Sect. 3.3 was then applied to the lake area to derive the cumulative daily lake volume over
this region over the 2016 to 2022 melt seasons (Fig. 7). The average maximum (peak) daily lake volume is $7.97\pm1.96\cdot10^{8}$ m$^{3}$
(one standard deviation) over all considered years. Most years tend to reach a fairly similar maximum daily lake volume, with
the exception of 2018, which was a particularly cold and dry year. Each melt season, however, exhibits distinct lake volume





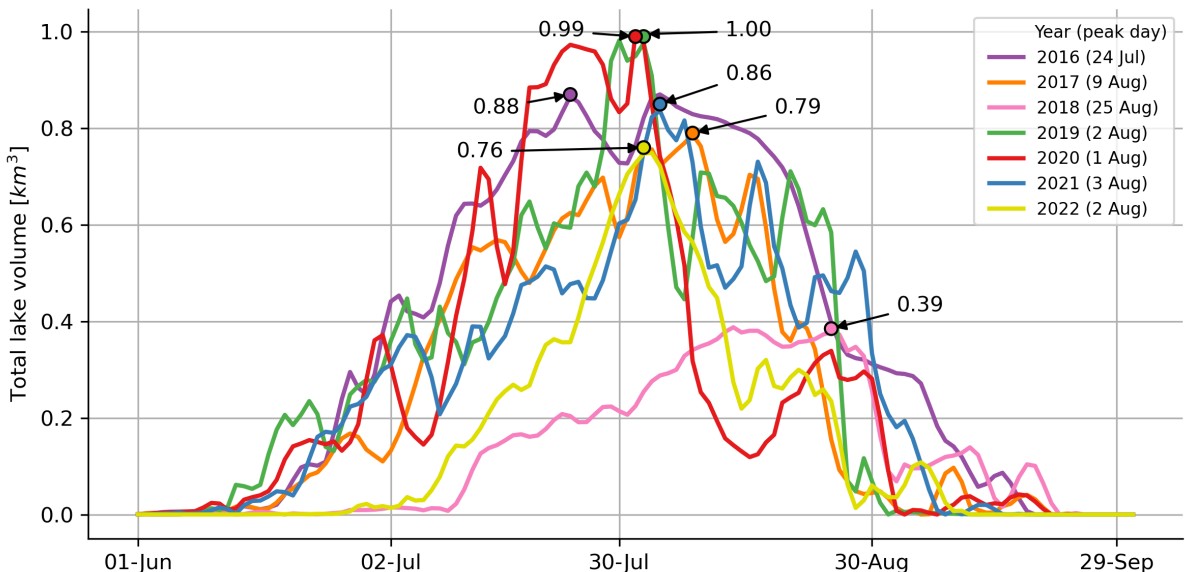

**Figure 7.** Daily lake volumes upstream of the 79NG and Zachariæ Isstrøm grounding lines throughout the 2016 to 2022 melt seasons. The maximum daily lake volume is marked by a dot for each year, with the corresponding date of the peak given in the legend.

accumulation and draining trends. For example, the 2020 and 2021 melt seasons have similar total daily lake volumes of around

$8.60 \cdot 10^8$ m$^3$; however, the 2020 melt season accumulates lake volume earlier and shows roughly double the meltwater volume on around July 25 than the year 2021. Additionally, in most years the peak lake volume is maintained for several days only; however, in 2016 the peak of $8.76 \cdot 10^8$ m$^3$ is maintained for a few weeks.

### 4.4 Tidal response of the 79NG

We now move in from surface processes to the dynamics of the 79NG. Comparing simulated and observed horizontal and ver-

tical displacements of the 79N over a tidal cycle revealed that the representation of the ice properties by a Maxwell viscoelastic rheology is required in the model, since a purely viscous material is not sufficient to describe the response of glacier ice to tidal forcing in a realistic way (Christmann et al., 2021). In addition, tidal forcing leads to an alteration of the subglacial water pressure which subsequently affects glacier motion due to changes in lubrication and, consequently, sliding speed. This effect is pronounced up to a distance of 10 km upstream from the grounding line, where the contribution of vertical shear to overall

motion is vanishing. Both viscous and elastic deformation is largest in the vicinity of topographic changes that initiate changes in stress. The elastic contribution is persistent and makes up 6-30% of the strain in our simulation experiment. Furthermore, the location of massive crevasse fields, that represent the solid nature of ice, are consistent with locations of high elastic strain.



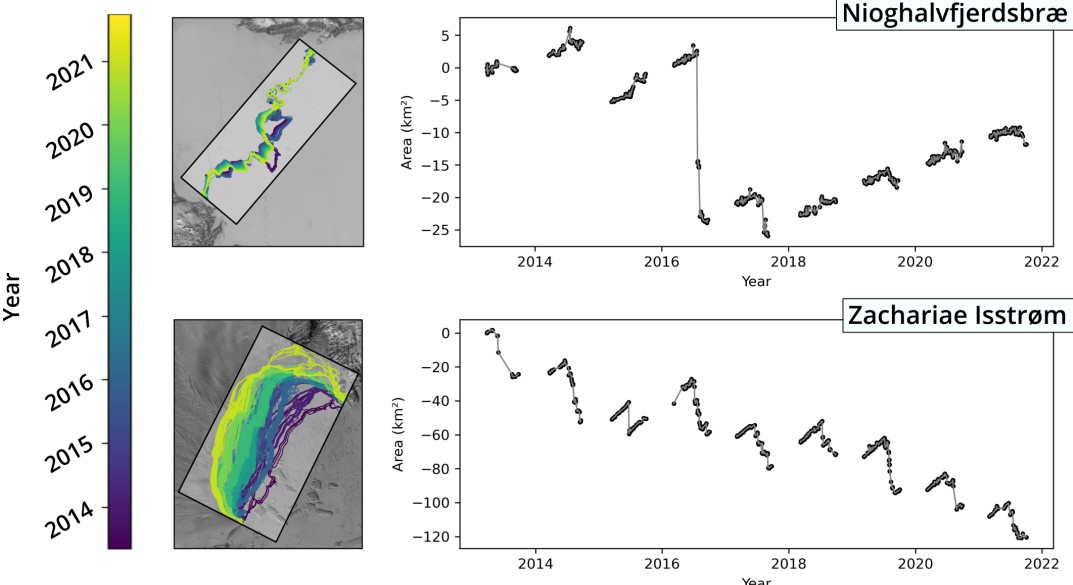

**Figure 8.** Time series of frontal positions generated by our deep neural network for 79NG and ZI. For both glaciers a satellite image (left), showing the color-coded calving front trajectories, and the corresponding time series (right) are depicted. Here, calving front positions are marked with a black dot and solid lines connect entries each year. Note that the different area scales (y-axis range) of the two right hand panels

## 4.5 Determination of frontal positions of 79NG and ZI

The analysis of Landsat imagery clearly shows that the calving regime of both glaciers is completely different, although they
are situated in direct vicinity. The frontal position of ZI exhibits a retreat of about 7 km from March 2013 to October 2021 with a distinct seasonal signal (Fig. 8, lower right panel). In contrast to that, the 79NG maintains its front more or less at the same position (upper right panel). However, events of advance followed by iceberg breakups can be detected. The frontal retreat of ZI is in accordance with its significant mass loss rate, which is about 3.5-times larger than that of 79NG (Sect. 4.7).

## 4.6 Sensitivity of the 79NG ice discharge to ice tongue extent

We investigated the impact of a possible disintegration of the floating tongue of the 79NG on the solid ice discharge based on a high-resolution setup of the ISSM model (Humbert et al., 2023). For the 79NG Mouginot et al. (2015) provide an annual mean estimate of ice discharge across the grounding line of $12.0 \pm 0.8$ Gt a$^{-1}$ for the period 2010-2015 based on remote sensing data. Our ISSM-based reference simulation ("init" simulation shown in Fig. 9), yields an equilibrium ice discharge of 11.9 Gt a$^{-1}$, which is comfortably within the error bars of the remote sensing-based estimate.
We found that an entire loss of the floating part approximately doubles the discharge of ice across the grounding line (Fig. 9). Comparing the small change in discharge detected by Mankoff et al. (2020b) between 1986-2020 with our simulations, we




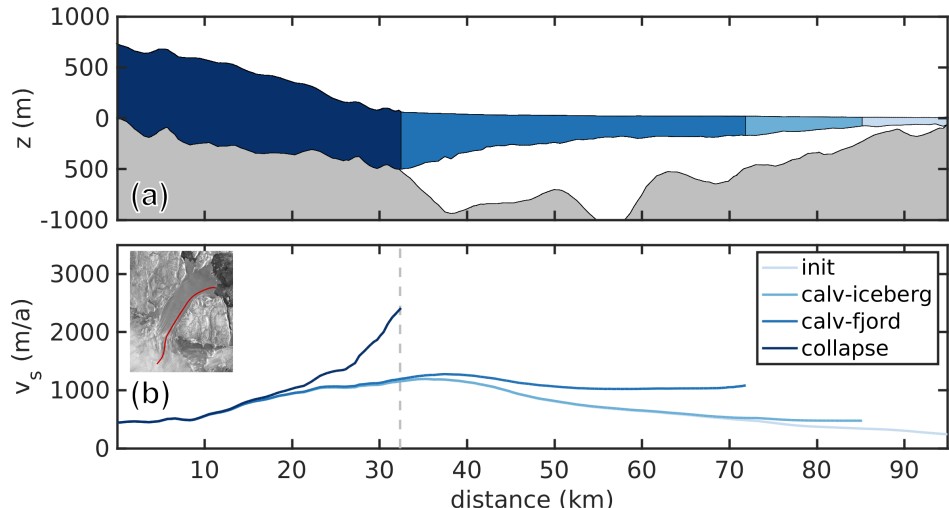

**Figure 9.** Simulation results along a south-eastern flowline as demonstration of the 3D ice flow simulation (see inset in (b) for the location) for the four experiments init, calv-iceberg, calv-fjord and a full collapse of the floating tongue (from light to dark blue, respectively). Panel (a) shows the geometry and panel (b) shows the simulated velocities $v_s$. The gray dashed line in (b) indicates the location of the grounding line.

conclude that, although the floating tongue of 79NG has thinned considerably (Mayer et al., 2018; Khan et al., 2022), the buttressing still prohibits major speed-up. Only for retreats reducing the length of the ice tongue to less than 40 km, the discharge starts to accelerate (experiments calv-fjord and collapse). The basal topography inland, with a steep step at a distance

of about 20 km upstream the grounding line, attenuates the effect of the loss of the floating tongue far inland.

## 4.7 Refined mass balance of the Greenland Ice Sheet

The combination of satellite gravimetry and altimetry yields a refined and improved estimate of the ice-mass change at a high resolution in the working area (Fig. 10). While the higher resolution is facilitated by satellite altimetry, satellite gravimetry constrains the mass balance estimate towards a physically more reliable solution, since GRACE/GRACE-FO directly senses

mass change. Further, the combination of the two methods maintains the spatial pattern of the altimetry input which is a significant improvement over the gravimetry-only estimate in terms of the spatial resolution and allocation of mass changes to single glaciers (Kappelsberger et al., 2021). A major advantage of the combination method is that the contribution of the peripheral glaciers is reliably considered which makes up more than 10% of the entire Greenland mass balance (Sect. 4.9). This is also important for obtaining a correct mass change distribution over the different catchment areas and regions of the

peripheral glaciers. The estimate from the altimetry-gravimetry combination for the time period 07/2010 to 06/2017 yields a mass-loss rate of -3.7±1.3 Gt a$^{-1}$ and -0.9 ± 0.4 Gt a$^{-1}$ for the drainage basins of ZI and 79NG, respectively. For comparison, for the same period the overall Greenland ice-mass trend ist -233±43 Gt a$^{-1}$, which is comparable to other recent estimates. 83% of ice-mass trend is contributed by the GrIS and 13% by peripheral glaciers.




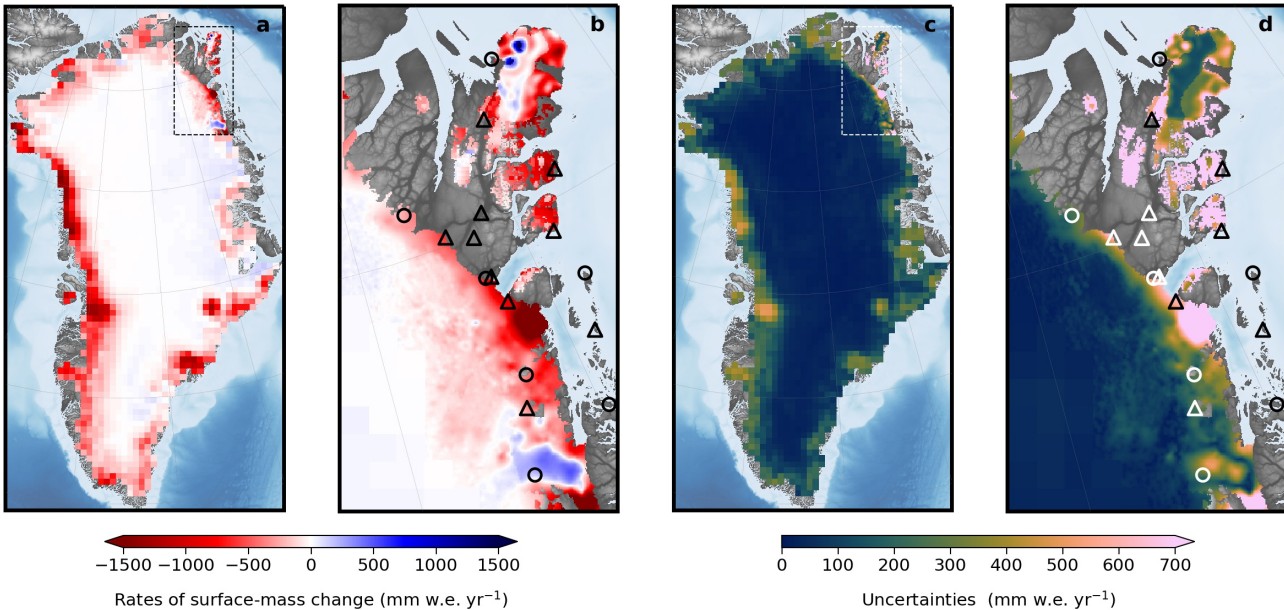

**Figure 10.** Surface-mass change rates from the combination of satellite gravimetry and satellite altimetry for entire Greenland (a) and the working area in north-east Greenland (b), and related uncertainties (c) and (d). Time period: 07/2010 - 06/2017. NG: 79 North Glacier, ZI: Zachariæ Isbræ, SN: Storstrømmen. Locations of the GNSS sites are given by triangles (TU Dresden) and circles (GNET), cf. also Figs. 3 and 4. Figure from Kappelsberger et al. (2021).

## 4.8 Bedrock displacement rates inferred from GNSS and the response of the solid Earth to present-day ice-mass changes

From the analysis of the GNSS measurements at the individual TUD and GNET sites (Fig. 3) we inferred bedrock displacement rates. Here, especially the uplift rates (Fig. 11) are of great relevance ranging between 5 mm a$^{-1}$ and a maximum of almost 9 mm a$^{-1}$ at western Lambert Land. The uplift rates contain both the long-term GIA effect as well as the elastic response due to present-day ice-mass change. The GNSS site at western Lambert Land is situated close to ZI. It features the largest contemporary ice-mass change in the region, which reflects in the largest observable elastic response in NE Greenland. Therefore, a new permanent GNSS site was set up at this location in July 2022 to continue the time series and resolve seasonal and interannual signal parts of the solid Earth response.

The elastic response of the solid Earth was calculated based on the refined present-day mass change estimate (Fig. 11). This calculation also accounts for load variations of the surrounding ocean as well as for mass redistributions on a global scale. These two effects can account for up to 10% of the entire elastic effect (Kappelsberger et al., 2021). The largest elastic displacement rate (13.8 mm a$^{-1}$) was obtained near the front of ZI (Fig. 11). The sum of the elastic displacements and those inferred from different GIA models could then be compared with the GNSS-inferred uplift rates. From that validation it turned





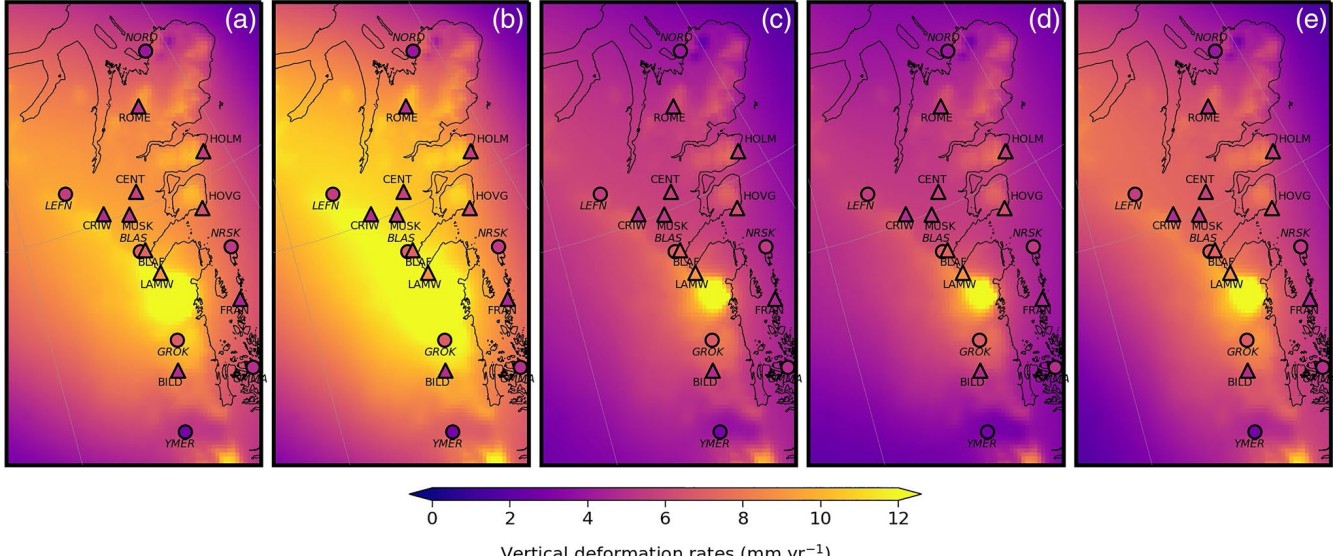

**Figure 11.** Total vertical displacement rates inferred from the sum of the elastic (instantaneous) response (calculated based on the mass-change rate from altimetry-gravimetry combination) and the viscoelastic (long-term) response predicted by different GIA models (a) A et al. (2012), (b) Peltier et al. (2015), (c) Caron et al. (2018), (d) Lecavalier et al. (2014), (e) Khan et al. (2016) (GNET). GNSS-inferred vertical displacement rates, inferred for the time period 06/2008 to 07/2017, are shown in the same color scale, for TU Dresden sites (triangles) and GNET sites (circles). Figure from Kappelsberger et al. (2021).

out that the GIA models by Caron et al. (2018); Lecavalier et al. (2014); Khan et al. (2016) are compatible with our results inferred from the combined mass change estimate and the GNSS measurements (Fig. 11). These three GIA models predict

a relatively small GIA signal in NE Greenland with rates at the GNSS sites ranging from 0.7 to 4.4 mm $a^{-1}$. While for the model by (Khan et al., 2016) a good agreement was expected since it is based on the GNET data, for the other two models the agreement is remarkable. Lecavalier et al. (2014) presented a regional model based on comprehensive records of past ice extent and relative sea-level (RSL) data. Caron et al. (2018) included only a few RSL data and no GNSS-inferred rates in Greenland at all. However, since the determination of GIA represents a highly non-unique problem, research into better constraining both

the ice-load history and the rheology of the Earth will remain a challenge which form the two primary quantities for GIA forward modeling.

### 4.9   Mass balance of peripheral glaciers in Northeast Greenland

The differences in the spatial resolution of the climate drivers (input data) play a major role in the simulated glacier and hydrologic outputs from OGGM (Sect. 3.9). The freshwater runoff contributions from FIIC were 11.40, 9.57, and 9.93 Gt

$a^{-1}$ when simulated using the CRU, ERA5, and NEGIS_WRF datasets as atmospheric boundary conditions, respectively, during 2014-2017 (Fig. 12d-f). Despite their different spatial resolutions, there were no significant statistical differences in



freshwater runoff contributions across datasets. However, during the mass balance calibration process within OGGM, the coarse resolution datasets (CRU and ERA5) were corrected using precipitation scaling factors (2.5 and 1.6, respectively). High-resolution precipitation based on NEGIS_WRF, by contrast, was sufficient to calibrate the mass balance without applying corrections, increasing confidence in these results.

The grounding lines of the ice shelf in the two basins of FIIC showed a slight advance during the 1996-2001 surge activity ("Svalbard-type"). Since 2001 the grounding lines of basins 2 and 3 have retreated by 2.2±1.3 km and 2.7±0.9 km (Möller et al., 2022), suggesting an ongoing mass loss in the area. This finding is consistent with the observations of Kappelsberger et al. (2021) (Fig. 12c). During the surges, ice discharge from basins 2 and 3 reached amounts of 0.145 and 0.119 Gt a$^{-1}$. Afterward, the discharged ice masses rapidly decreased to a distinctly lower level of just 10% (basin 2) and 4% (basin 3) of their maximum values. OGGM simulations revealed a total mass loss from FIIC of 2.8±0.7 Gt a$^{-1}$ from 2010 to 2019.

The peripheral glaciers in NE Greenland underwent a significant mass change, with a calculated mass loss of 8.1±4.3 Gt a$^{-1}$ between 2010 and 2019. These simulated mass changes agree with the estimates of Khan et al. (2022). The peripheral glaciers in NE Greenland will lose significant volume in the future. The remaining volume is predicted to be 78±4% for SSP126 and 61±9% for SSP585 by 2100 compared to 2019 (Fig. 12g). Based on the future projections, the total mass loss from the peripheral glaciers of NE Greenland is 5.15±0.88, 6.44±1.19, 7.94±1.73, and 9.04±2.16 mm sea level equivalent (SLE) under SSP126, SSP245, SSP370, and SSP585, respectively, by 2100 compared to the mass in 2019 (Fig. 12h). The values in parenthesis represent the actual sea level rise (dotted lines in Fig. 12h) after excluding the mass loss below water level. The uncertainty in the projections was higher for high-emission scenarios.

Projections of freshwater contributions from peripheral glaciers into the ocean are essential for understanding how the interactions between glaciers and the ocean will evolve in a warming climate. Increased meltwater will eventually end up in the ocean, altering the density distribution within the fjords and, thereby, fjord circulations and submarine melt rates (Hopwood et al., 2020; Henson et al., 2022). Total liquid freshwater contributions from NE Greenland (excluding submarine melt and calving fluxes) to the ocean are projected to be 3922±315 Gt (SSP126) and 5576±826 Gt (SSP585) during 2020-2100. Of specific interest is the year of peak water, which is the point in time when total annual runoff from a glacier reaches its temporal maximum under a given climate scenario. In high emission scenarios, this point in time is usually reached later than in low emission scenarios, since the excess melt can continue to increase over the rest of the 21st century. For the lower emission scenarios, the glacier would start to approach a new equilibrium within the 21st century, which reduces the runoff. The peak waters are projected to occur in the years 2053±22 (51.8±16.6 Gt a$^{-1}$), 2070±19 (54.5±12.9 Gt a$^{-1}$), 2082±17 (80.4±17.9 Gt a$^{-1}$), and 2080±19 (79.2±19.3 Gt a$^{-1}$) under SSP126, SSP245, SSP370, and SSP585, respectively (Fig. 12i).



**Figure 12.** (a) Greenland peripheral glaciers and new outlines of Flade Isblink Ice Cap (FIIC). Glaciers are highlighted in different colors for different subregions. FIIC new outlines highlighted in cyan are active marine-terminating glaciers. (b) thick black and white outline represents the delineation of basins 2 and 3 that was drawn based on 2019 Sentinel-1 surface ice flow velocities (color grid) and directions (arrows) (Möller et al., 2022) (c) Mass balance based on altimetry and gravimetry obtained from Kappelsberger et al. (2021). (d-f) Freshwater runoff, simulated using different spatial resolution forcing datasets. (g) Projected remaining volume of peripheral glaciers in NE Greenland by 2100 (for low and high emission scenarios) compared to the volume in 2019. (h) Projected total glacier mass loss up to 2100 (10 GCMs: mean ± 1 SD) from NE Greenland peripheral glaciers compared to 2019. The dotted lines represent the mean sea level rise after subtracting the mass loss below water level under SSPs. (i) Projected years of peak water (mean ± 1 SD) for the freshwater runoff contributions from peripheral glaciers of NE Greenland under SSP126, SSP245, SSP370, and SSP585 (10 GCMs: mean ± 1 SD).



## 4.10  New insights of ocean circulation in the cavity by observations and models

After having discussed ice sheet and glacier processes relevant for the mass balance of the 79NG system and adjacent peripheral glaciers, we move on to ocean impacts (i.e. the ocean-driven melt in Sect. 4.10 to 4.12 and impacts of the melt water on the ocean in Sect. 4.13). The mooring-based time series of ocean velocity, temperature, and salinity near the calving front of 79NG
from 2016 to 2017 (location given as green dots in Figs. 3 and 4) indicate the existence of a year-round bottom-intensified inflow of warm AIW into the cavity through a narrow channel across a chain of small islands to the calving front, suggesting a year-round ocean-driven basal melt to take place (Schaffer et al., 2020). This consistent inflow is confirmed by further hydrographic observations at the calving front until 2022 (Fig. 13). The combination of the observations with a bathymetric survey reveals that this inflow is constrained via hydraulic control by a topographic sill located several kilometers upstream of the calving
front (for process see Fig. 2), meaning that the height of the warm and saline AIW layer above the submarine sill controls the magnitude of AIW volume (and heat) transport into the cavity of the 79NG. As a consequence, AIW enters the cavity as a bottom-intensified dense water plume (see velocity and temperature observations downstream of the sill at the calving front in Fig. 13). Temperature profiles at the neighboring ZI suggest that ocean heat transport here is similarly controlled by a near-glacier sill (Schaffer et al., 2020). Near-glacier sill-controlled ocean heat transport thus plays a crucial role for glacier stability.
Based on the same mooring data, (von Albedyll et al., 2021) found that the deep AIW inflow is connected to a shallow outflow of modified (cooled and freshened) AIW on time scales exceeding one month, with exchange flow variability being intensified in wintertime. The extent to which the intraseasonal variability is relevant for modulating melt rates is unclear, as the residence time of seawater in the cavity exceeds 6 months.

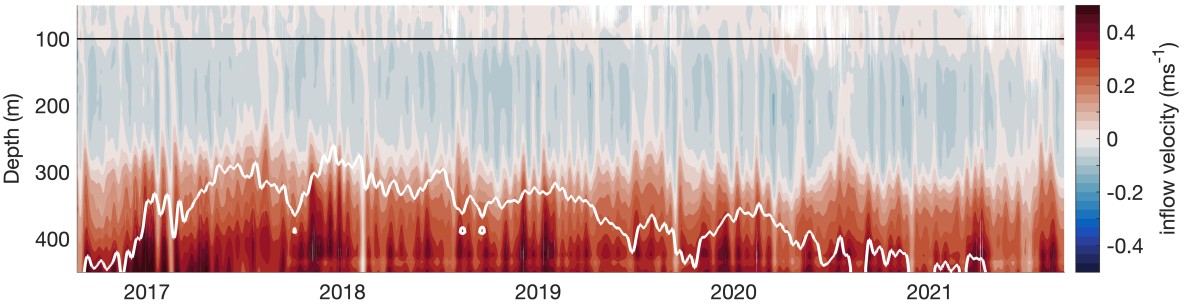

**Figure 13.** Velocities based on a mooring at the calving front of the 79NG, where positive/red (negative/blue) velocities are the inflow (outflow) towards (away from) the cavity. The white line indicates the depth of the 1.2° C isotherm. The depth of the glacier base at the calving front is 100 m, above which there is no data. Data is averaged between August 2016 and September 2021 and low-pass filtered using a window of 2 weeks.

As oceanographic observations are mostly constrained to the calving front and open ocean, we make use of ocean models
to study the ocean circulation inside the glacier cavity. The 2D fjord model (2D in the x-z-plane) features both an along-fjord bathymetry and an ice tongue geometry similar to that of the 79NG (Reinert et al., 2023). The 2D fjord model reveals





a meltwater plume that forms on the underside of the floating ice tongue, initiated by subglacial runoff discharged at the grounding line. It rises along the ice tongue, since it is fresher and thus lighter than the surrounding AIW in the cavity (see schematic in Fig. 2). Being equipped with adaptive vertical coordinates (Sect. 3.13) and featuring 100 vertical layers, the model is ideally suited to simulate the small scale turbulent exchange processes between the meltwater plume, the ice base above and the AIW layer below. The plume causes basal melt of the ice tongue due to friction-imposed ocean heat flux at the ice–ocean interface. The meltwater is absorbed by the plume and increases its buoyancy. On the other hand, the plume becomes less buoyant due to the entrainment of AIW from below (Burchard et al., 2022). Eventually, the plume water reaches the same density as the surrounding AIW. It detaches from the ice tongue, propagates horizontally away from the ice (Reinert et al., 2023), and transports glacially modified (i.e. cooled and freshened) water out of the fjord (Fig. 2) at mid-depths. This finding is supported by our observations of the subsurface distribution of helium and neon near the calving front (Fig. 17c; Sect. 4.13).

Our 3D ocean-sea ice model, featuring a horizonatal mesh resolution of 700 m in the cavity and its surrounding, reveals that the warm and salty AIW enters the cavity through its deepest channel as a bottom-intensified gravity current (Wekerle et al., 2024), in agreement with Schaffer et al. (2020). The warm AIW then circulates cyclonically around the cavity. The year-round inflow of AIW results in ocean temperatures above the freshwater freezing point in the cavity, leading to basal melt (as in the 2D fjord model). Both basal melt water and subglacial discharge which enters the cavity in the summer months across the grounding line, sustain a meltwater plume which entrains ambient water as it rises. Driven by buoyancy and deflected by the Coriolis force, the plume accelerates along the southern flank of the cavity and transports the meltwater out of the cavity. The ocean circulation inside the cavity is thus clearly affected by the Coriolis force, deviating strongly from a two-dimensional estuarine circulation. The basal melt rates computed by our model simulations are discussed in Sect. 4.11 below.

### 4.11  Basal melt of the floating tongue of the 79NG

The heat transport associated with the flow of AIW into the cavity leads to year-round melt at the base of the floating glacier tongue (Schaffer et al., 2020). It is the major contribution to the observed thinning of the ice tongue of the 79NG (Wilson et al., 2017; Mayer et al., 2018). Time series measurements of basal melt in close vicinity of the grounding line were carried out using ApRES devices (Zeising et al., 2023). The authors found that the bottom side of the 79NG's floating tongue has a relatively rough surface, so that the ApRES measurements of basal melt do not only record nadir reflections but also side reflections, which affects the ability to constrain the melt rates. Fig. 14 presents a timeseries of basal melt rates inferred from the ApRES measurements and reveals strong temporal and spatial (the difference between the median and 95% quantile) variability, with no clear seasonal cycle. The median melt rates increased from 37 m a$^{-1}$ in October 2016 to 58m a$^{-1}$ at the end of 2017, though maximum values exceeded 120 m a$^{-1}$ (95% quantile) recorded 5.5 km downstream from the grounding line. From early 2018 onward melt rates then steadily decreased to January 2020, reducing by over 50% of the 2017 peak, with reduced spatial variability. The reduction coincided with the observed decline of the AIW temperature near the calving front (Fig. 14), possibly suggesting ocean temperatures as a driver of the melt rate changes (see also Sect. 4.12).

In Fig. 15 we present maps of basal melt rates derived from different models and from remote sensing data. A common feature is the increased melt rate close to the grounding line, and its concentration in the southern part. Based on satellite



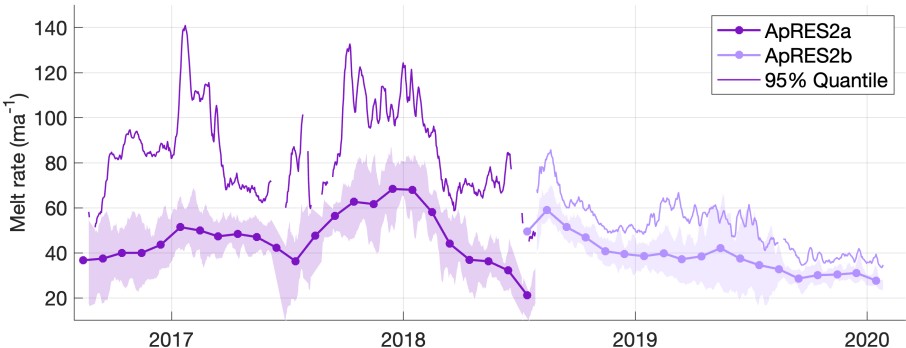

**Figure 14.** Basal melt rate time series of the ApRES2a and ApRES2b. The shaded area marks the range between the 25% and 75% quantile about the median (dotted line), and the solid line indicates the 95% quantile. Note that the ApRES drifted with the ice away from the grounding line over time and was relocated to the original starting point in July 2018.

imagery collected between 2011 and 2015, Wilson et al. (2017) estimated an area-averaged melt rate of 7.4 m a$^{-1}$ (assuming an ice-shelf area of 1600 km$^2$), with highest values of 50–60 m a$^{-1}$ close to the grounding line. In the 3D fjord model (Fig. 15b), we obtain area-averaged melt rates of 10.1 m a$^{-1}$, close to the value of 10.4 ± 3.1 m a$^{-1}$ derived from mooring-based observations at the calving front for the period of summer 2016 to summer 2017 (Schaffer et al., 2020). The 2D fjord

model produces an area-averaged melt rate of 12.3 m a$^{-1}$. The global 3D ocean–sea ice model (Fig. 15c) simulates a mean melt rate of 10.3 m a$^{-1}$. The satellite-derived melt rates by Wilson et al. (2017) (Fig. 15d) thus yield a lower mean melt rate than our estimates based on numerical models or observed hydrography. The satellite retrievals assume a hydrostatic equilibrium over the entire floating ice tongue, which may not be well justified close to the grounding line, where the highest melt rates occur. Basal melt rates of 8.6 ± 1.4 m a$^{-1}$, i.e. lower than our other estimates but still higher than the numbers suggested by Wilson

et al. (2017), were obtained by Huhn et al. (2021) based on measurements of helium and neon on the NE Greenland Shelf and close to the 79NG in summer 2016. The above melt rates in units m a$^{-1}$ can be converted to Gt a$^{-1}$ and a meltwater flux in mSv by multiplication of 1.7 and 0.053, respectively. Schaffer et al. (2020) estimate the subglacial discharge of freshwater into the cavity to amount to 11% of the basal meltwater flux, which is in rough agreement with corresponding estimates based on the 3D fjord and ocean-sea ice models. To put the above numbers of basal melt into perspective of ice discharge across the

grounding line, Mouginot et al. (2015) estimate of 12.0 ± 0.8 Gt a$^{-1}$ for the 79NG can be translated into a "vertical growth rate" of the ice tongue of 7 m a$^{-1}$ (obtained by division by 1.7; see above). This is clearly less than any of the melt rate estimates above, confirming the observed, decadal thinning of the floating ice tongue (see also Wilson et al. (2017)).

The sensitivity of basal melt rates to ocean temperature and subglacial discharge was tested with both the 2D fjord model and the 3D ocean-sea ice model (Table 1). Increasing the water temperature by 0.5° C and 1.0° C – representing realistic ranges of interannual to decadal changes (e.g. Schaffer et al. (2020) and Fig. 10) – leads to an increase in basal melt by 26% and 52%,

respectively in the 2D fjord model. A doubling of subglacial discharge – corresponding to the difference between a year with



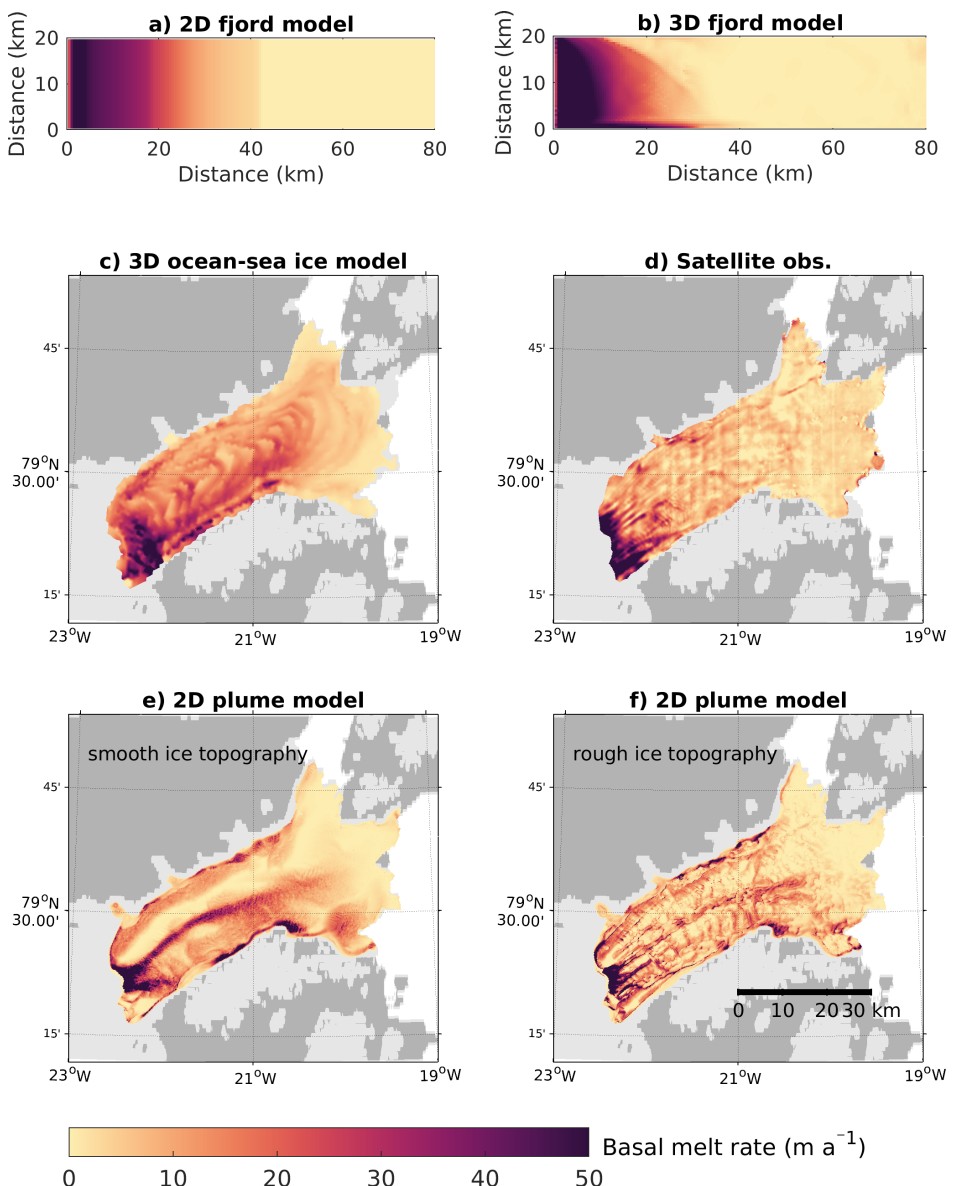

**Figure 15.** Basal melt rate (m a$^{-1}$) of the 79NG ice tongue from a) the 2D fjord model (Reinert et al., 2023), (b) the 3D fjord model, (c) the 3D ocean–sea ice model (FESOM2.1, Wekerle et al. (2024)), d) satellite observations (Wilson et al., 2017), and the 2D-plume model (Mohammadi-Aragh and Burchard, 2024) using (e) a smooth ice base topography and (f) a rough ice base topography. The gray and light gray background in (c)-(f) shows bare land and grounded ice, respectively.

average and one with high discharge (see Fig. 16 for a discharge time series) – increases the basal melt by 16%. From an ensemble of seven sensitivity experiments, Reinert et al. (2023) concluded that the basal melt rate can be best described by a function that depends on the square root of subglacial discharge and is around 7 m a$^{-1}$ in the absence of discharge. These



findings are consistent with results obtained with the global 3D ocean–sea ice model. Sensitivity experiments with reduced and enhanced subglacial discharge confirm the square-root relationship with basal melt (Wekerle et al., 2024). Regarding ocean temperatures, we find that mean basal melt rates follow a quadratic function of AIW inflow temperature in the 3D ocean-sea ice model. Moreover, the sensitivity of basal melt to the roughness of the ice base has been explored with the 2D plume model. An experiment featuring a smooth ice base geometry revealed a somewhat higher area-averaged melt rate of 9.2 m a$^{-1}$ compared

to that resulting from the rough geometry (8.1 m a$^{-1}$; compare Figs. 15e,f). The latter features a clear imprint of the locations of the subglacial channels on the distribution of melt rates across the tongue (Fig. 15f).

### 4.12   Origin and pathways of Atlantic Intermediate Water on the shelf of NE Greenland

Based on historical hydrographic data, Schaffer et al. (2017) demonstrated that the AIW present near the 79NG was transported by an anticyclonic circulation on the NE Greenland shelf, with Norske Trough (Fig. 3) representing the inflow pathway. From

the 2014–2016 mooring time series of temperature and velocity in central Norske Trough, Münchow et al. (2020) were able to identify a subsurface boundary current transporting about 0.27 Sv of AIW from Fram Strait towards the inner shelf. The intraseasonal variability of subsurface the AIW transport can be explained by topographic Rossby waves, generated by wind stress curl forcing on the Greenland shelf break. von Albedyll et al. (2021) showed the observed AIW transport in Norske Trough to co-vary with the AIW inflow into the cavity of the 79NG, however, with no significant lag, suggesting fast, wave-

like propagation of these fluctuations. Finally, Lindeman et al. (2020) observed that temperature anomalies obtained from the same (2016-2017) mooring at the calving front were rapidly transported into the cavity. In addition to a high degree of connectivity of AIW transports toward the 79NG on intraseasonal time scales, based on the 3D ocean-sea ice model covering the time period 1970-2021 , McPherson et al. (2023) found the interannual variability of the AIW on the continental shelf of NE Greenland to be driven primarily by the westward advection of Atlantic Water temperature from the WSC across Fram Strait.

The authors show that both temperature anomalies in the WSC and wind-driven anomalies of the ocean circulation in Fram Strait contribute significantly to the AIW temperature variability on the shelf. The 3D ocean simulation also shows that not the AIW temperature on the NE Greenland shelf but also the basal melt rates of the 79NG are strongly connected to the upstream ocean conditions (Fig. 16; Wekerle et al. (2024)). Simulated annual mean basal melt rates are significantly correlated with the inflow temperature at the calving front (r=0.85), whereas the correlation with the subglacial discharge across the grounding

line is not significant (r=0.22).

     Hydrographic data collected since the 1980s in the inflow pathway of the AIW roughly 100 km upstream of the sill at the 79NG calving front suggests that the thickness of the AIW layer on the continental shelf of NE Greenland has significantly increased over the past two decades. Given that the AIW inflow into the cavity is hydraulically controlled, this implies an increase in the basal melt rate of the 79NG (Schaffer et al., 2020). In agreement with the observations, the 3D ocean-sea ice

simulation revealed an increase in the AIW temperature at the calving front of the 79NG with a trend of 0.19° C per decade during the time period 1970-2021 (Fig. 16). The increasing AIW temperatures are strongly reflected in simulated basal melt rates (Fig. 16), which increase by 1.38 m a$^{-1}$ per decade. Wekerle et al. (2024) conclude that the temporal evolution of Atlantic Water in Fram Strait is the main driver of changes in basal melt rates of the 79NG over the past 50 years.



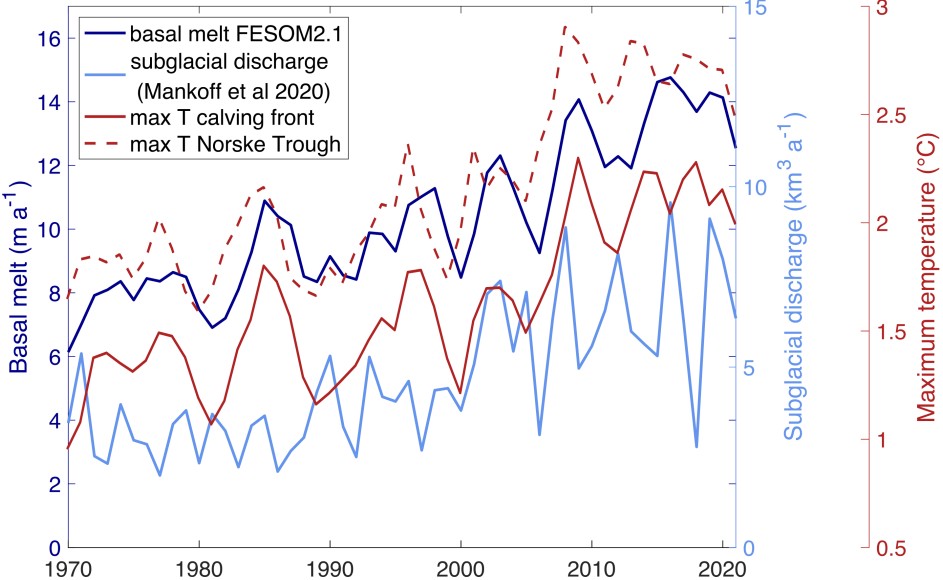

**Figure 16.** Simulated annual mean basal melt rate (dark blue line) and maximum AIW temperature at the calving front (red line) and in central Norske Trough (red dashed line) from the sea ice-ocean model FESOM2.1, and subglacial discharge from Mankoff et al. (2020a) (light blue). Figure adapted from Wekerle et al. (2024).

### 4.13 Spreading of meltwater around Greenland

Based on a year-long record of moorings, von Albedyll et al. (2021) found that around half of the outflow of glacially modified AIW from the 79NG was exported through Dijmphna Sund, and the other half through three exchange gateways along the main calving front. By measuring noble gases, the basal meltwater contained in the modified AIW can be traced on the NE Greenland shelf (Huhn et al. (2021), Fig. 17b/c). There it is present with highest concentrations of basal melt water between roughly 100 and 200 m, confirming the subsurface, density-equilibrated outflow of glacially modified waters from the cavity

simulated by the 2D fjord model (and the 3D ocean-sea ice model). The basal melt water fraction dilutes from concentrations of 1.8% at the calving front of the 79NG to non-detectable values ($< 0.1\%$) near the shelf break This suggests that a sizable meltwater impact on the ocean circulation should be restricted to the inner shelf. It also suggests that the basal melt water signal observed more than 2000 km downtream in the East Greenland Current close to Cap Farvel (Rhein et al., 2018) at the southern tip of Greenland (Fig. 1) does not originate from the 79NG, but from fjords further south.



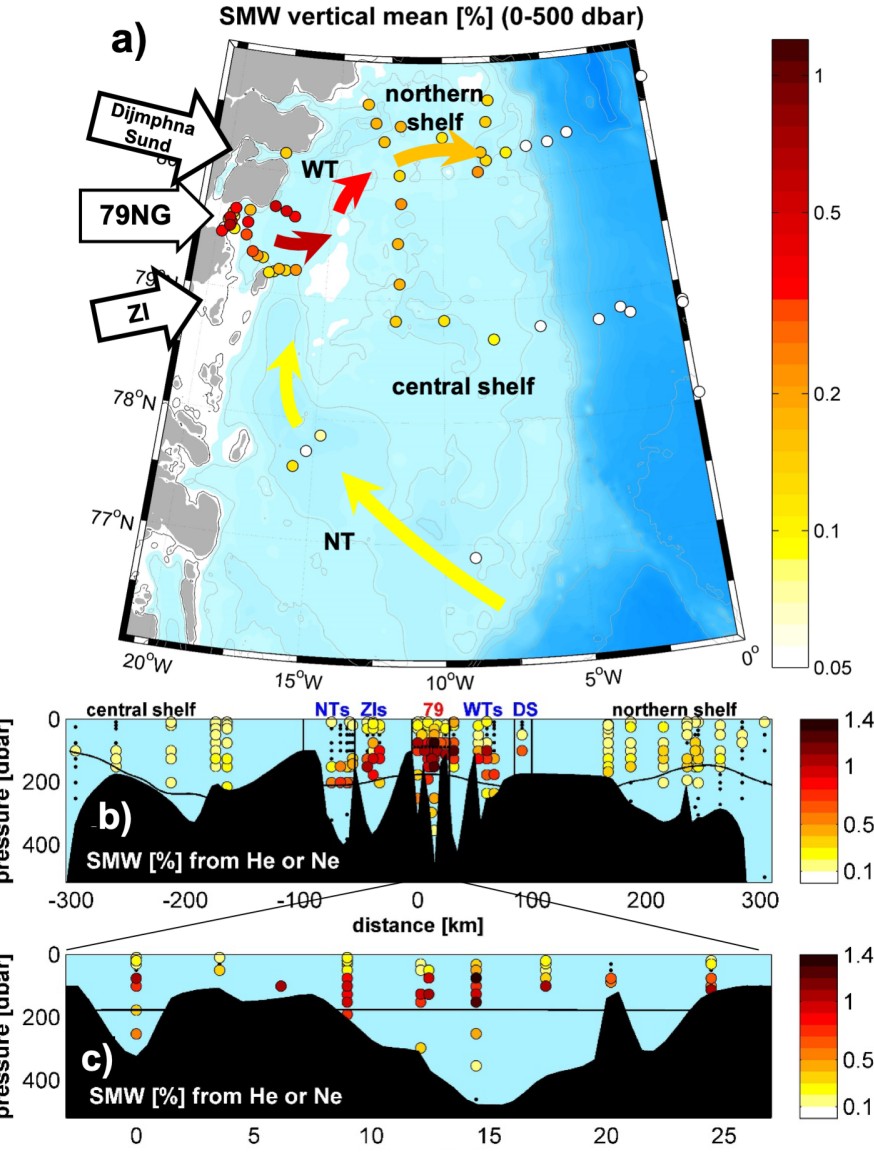

**Figure 17.** (a) Map of the NE Greenland continental shelf with noble gas based vertically averaged basal meltwater fractions from the uppermost 500 dbar (data from PS100, 2016). (b) Section with noble gas based basal meltwater fractions, following a line from Fram Strait across the slope onto the central shelf (~79° N) towards "NT" (Norske Trough), along the 79NG calving front, "WT" (Westwind Trough), and northern shelf (80° N) ending in Fram Strait. (c) Zoom into the same section along the 79NG calving front. Figure adaped from Huhn et al. (2021).

Simulations using the second setup of the 3D ocean-sea ice model (FESOM1.4; Sect. 3.13) allowed us to study the spreading of glacial meltwater on a larger scale (Stolzenberger et al., 2022), focusing not only on the 79NG as a meltwater source, but on the entire Greenland ice sheet. The meltwater runoff has a clearly detectable impact on the salinity and temperature distributions




in the ocean surrounding Greenland. At the east coast of Greenland, we find a near-coastal band featuring a pronounced drop in sea surface salinity exceeding 0.5 when considering a realistic rate of Greenland freshwater runoff compared to the reference

run without runoff (Fig. 18). This suggests any anomalous circulation response to the freshening (via geostrophy) to be limited to the coastal current system, and not to extend to the EGC branches at the mid-shelf and shelf break. At the west coast of Greenland, the negative anomaly in salinity is present thoughout Baffin Bay (not shown). Moreover, a warm anomaly of up to 0.5° C occurs along the west coast. As we only modified the meltwater runoff between the two simulation, temperature variations are entirely due to the dynamic response of freshwater input.

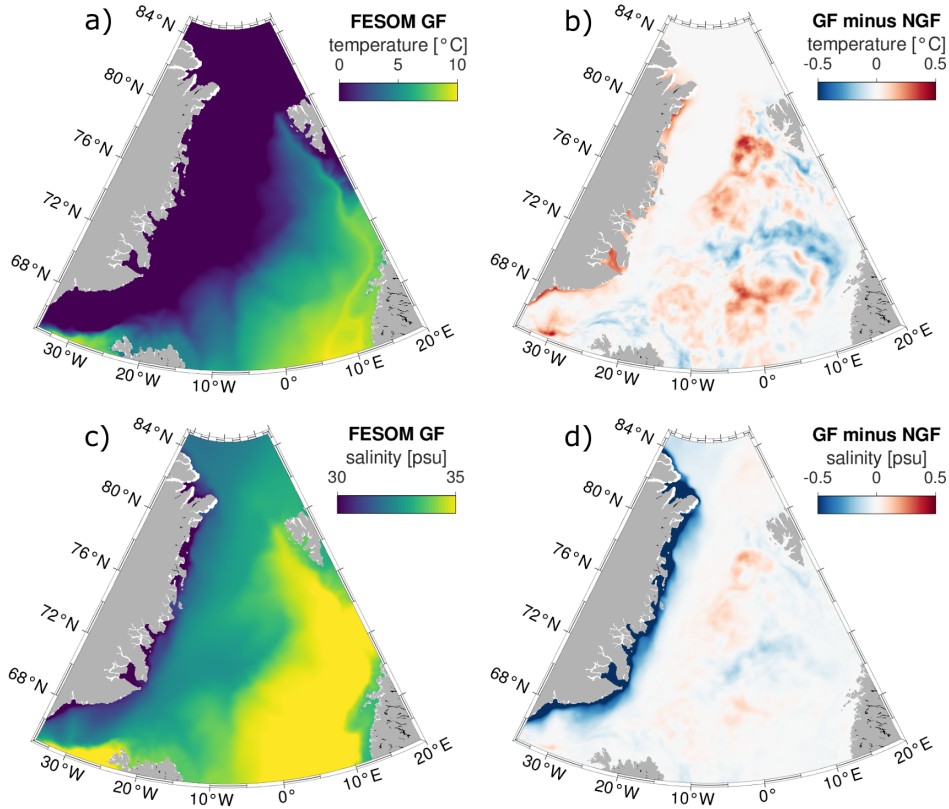

**Figure 18.** Surface temperature (a,b) and surface salinity (c,d) in the Greenland freshwater experiment (GF, left) and difference between the GF experiment and an experiment without Greenland freshwater (NGF). Blue shadings represent cooling (for temperature) and freshening (for salinity) effects due to GF. The time period is 1993–2016.

# 5 Summary and synthesis

Here we summarize our main findings with the goal of piecing together the most important estimates relevant for the mass balance of the 79NG ice–ocean system made in the framework of the GROCE project. Fig. 19 and Table 1 contain the condensed information. Atmospheric data reveal that the NEGIS area has experienced a multidecadal surface air temperature increase of



0.8° C per decade (1979–2017). Accordingly, our 2014–2018 results show that the 79NG clearly was in mass-loss mode with a

negative mean specific CMB (near-surface mass balance) of -744 mm a$^{-1}$; year-to-year variations, however, are strong and can only be understood from the combination of summer air temperature and precipitation during the previous winter, in particular, the solid precipitation fraction, that affects the albedo. A new time series of supraglacial lake volumes for 2016–2022 shows that the annual mean CMB and annual maximum supraglacial lake volume tend to co-vary. Maximum volumes are found in the late July–early August period. The year-to-year variability of the maximum lake volumes, in turn, is strongly tied to

the seasonal weather conditions; yet these volumes just vary between ±15% of the 7-year mean value (when excluding the exceptionally cold year of 2018).

From an integrated ocean–glacier study perspective, our approach has provided a high spatial resolution assessment of both the characterization of the interannual variability and a process-based understanding of the surface processes related to the 79NG. The next challenge would be to apply the surface melt water as a boundary condition to a hydrological model (such as

CUAS) of the 79NG interior, such that the impact on the glacier dynamics (sliding) and subglacial discharge-driven basal melt at the underside of the ice could be studied in a seamless way.

Regarding dynamics of the 79NG, we find that ice flow velocities driven by tidal forcing of the floating ice tongue propagate up to 10 km upstream of the grounding line, based on viscoelastic simulations. Satellite observations reveal that for 79NG, frontal positions have been relatively stable over the 2013–2021 period. In contrast, ZI exhibits a remarkable retreat of its front

of more than 7 km amounting to a loss of 120 km$^2$ of ice-covered area over the same time span. This result can be reconciled with the fact that the ice mass loss within the ZI drainage basin – inferred from remote sensing – is four times larger than that within the 79NG drainage basin for the 2013–2017 period. The retreat of the ZI also coincides with an increase in ice discharge by 50% from 2000 to 2015 (Mouginot et al., 2015), with no trend present in the two preceding decades. The pronounced decline of the ZI is also reflected in the GIA response of the solid earth in NE Greenland, obtained from different modeling

approaches supported by station-based GNSS observations. The latter reveal vertical ground displacement rates to display maximum values at the calving front of the declining ZI, which clearly exceed those at the calving front of the 79NG. While it has been found that the floating ice tongue of the 79NG has been thinning significantly over the past decades – including results obtained within GROCE – the important question still remains how susceptible the current ice flow regime of the glacier is to a future change of the ice tongue. In the period from 2000 to 2015 no significant trend in ice discharge was observed (Mouginot

et al., 2015). Using numerical model simulations we find that only in an extreme scenario, where about half of the length of the floating ice tongue is artificially removed, the 79NG enters a mode of a strongly increased grounded ice discharge towards the ocean. In less extreme scenarios, buttressing still prohibits a major speed-up of 79NG, implying that the glacier is unlikely to assume such a mode in the next few decades.



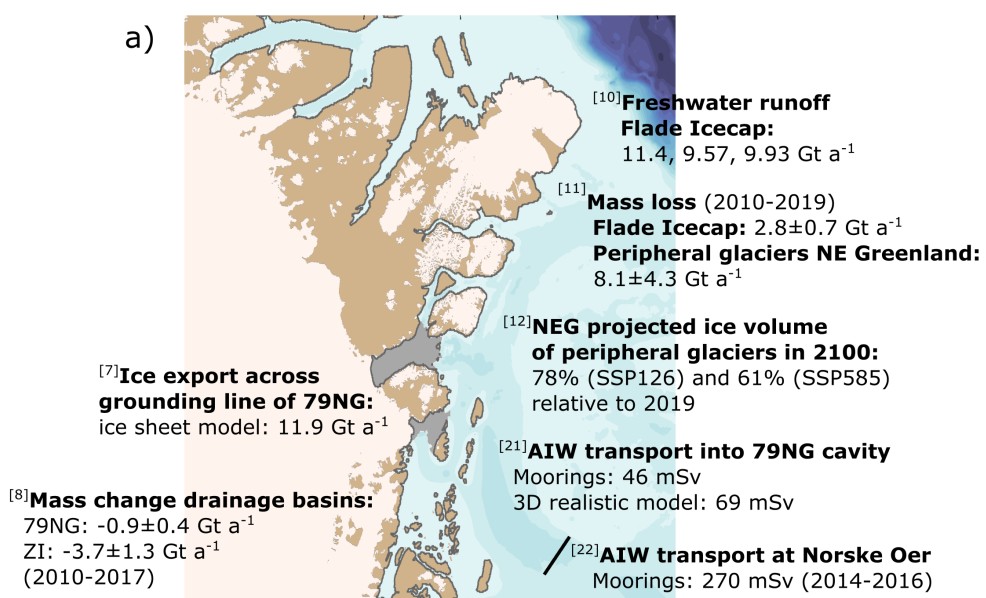

a)

[10]**Freshwater runoff Flade Icecap:**
11.4, 9.57, 9.93 Gt a$^{-1}$

[11]**Mass loss** (2010-2019)
**Flade Icecap:** 2.8±0.7 Gt a$^{-1}$
**Peripheral glaciers NE Greenland:**
8.1±4.3 Gt a$^{-1}$

[12]**NEG projected ice volume of peripheral glaciers in 2100:**
78% (SSP126) and 61% (SSP585) relative to 2019

[21]**AIW transport into 79NG cavity**
Moorings: 46 mSv
3D realistic model: 69 mSv

[22]**AIW transport at Norske Oer**
Moorings: 270 mSv (2014-2016)

[7]**Ice export across grounding line of 79NG:**
ice sheet model: 11.9 Gt a$^{-1}$

[8]**Mass change drainage basins:**
79NG: -0.9±0.4 Gt a$^{-1}$
ZI: -3.7±1.3 Gt a$^{-1}$
(2010-2017)

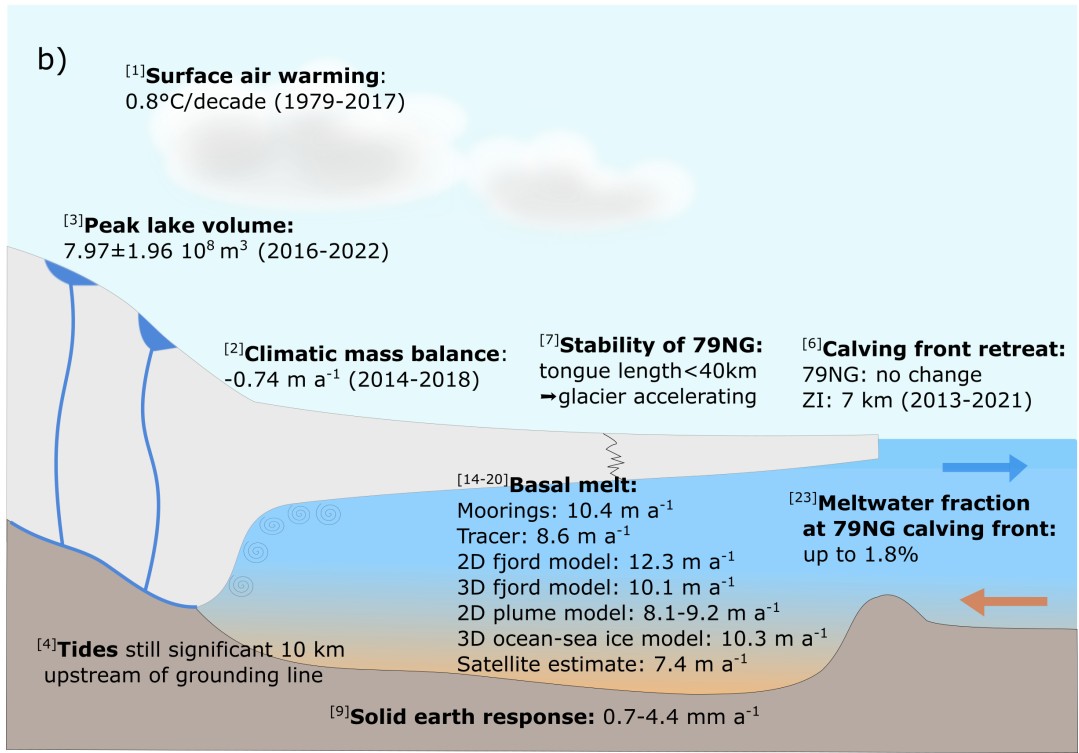

b)

[1]**Surface air warming**:
0.8°C/decade (1979-2017)

[3]**Peak lake volume:**
7.97±1.96 10$^8$ m$^3$ (2016-2022)

[2]**Climatic mass balance**:
-0.74 m a$^{-1}$ (2014-2018)

[7]**Stability of 79NG:**
tongue length<40km
➡glacier accelerating

[6]**Calving front retreat:**
79NG: no change
ZI: 7 km (2013-2021)

[14-20]**Basal melt:**
Moorings: 10.4 m a$^{-1}$
Tracer: 8.6 m a$^{-1}$
2D fjord model: 12.3 m a$^{-1}$
3D fjord model: 10.1 m a$^{-1}$
2D plume model: 8.1-9.2 m a$^{-1}$
3D ocean-sea ice model: 10.3 m a$^{-1}$
Satellite estimate: 7.4 m a$^{-1}$

[23]**Meltwater fraction at 79NG calving front:**
up to 1.8%

[4]**Tides** still significant 10 km upstream of grounding line

[9]**Solid earth response:** 0.7-4.4 mm a$^{-1}$

**Figure 19.** Essence of mass balance-related estimates of the GROCE project. Panel (a) shows the far-field estimates, while panel (b) displays the estimates related to the 79NG. The numbers given in brackets preceding each entry refer to the corresponding entry in Table 1, which gives more detailed information.





Our estimates reveal that the recent mass loss of FIIC (2010-2019) amounts to as much as 70% of the ZI (2010-2017) value.
Regarding the annual freshwater runoff from FIIC, our estimates range between 9.6 and 11.4 Gt $a^{-1}$, with 9.9 Gt $a^{-1}$ being the best guess; in this regard, the high-resolution (NEGIS-WRF) forcing data improved the calibration of mass balance without requiring additional corrections. Based on our future projections, a considerable increase in mass loss from the peripheral glaciers of NE Greenland is anticipated by the end of the 21st century relative to the mass in 2019. The mass loss will likely contribute to a sea level rise between 4 and 8 mm, contingent upon the specific emission scenarios (as illustrated in. Recent
investigations reveal that the mass loss from peripheral glaciers in Northern Greenland has exceeded the combined total of all other regions in Greenland over the past several decades (Khan et al., 2022). It is projected that, in the future, the NE region will maintain its status as the predominant contributor to the total liquid freshwater, solid ice calving, and sea-level rise among Greenland's peripheral glaciers (Shafeeque et al., 2023).

Basal melt rates of the 79NG – another primary focus of the GROCE project – were derived from observational and numerical
modeling approaches. Our observations from a year-long mooring-based time series of ocean heat transport allowed us to estimate the mean basal melt rate to be 10.4±3.1 m $a^{-1}$, while the observed tracer-based approach yielded 8.6±1.4 m $a^{-1}$. These two values represent bulk estimates of the melt rates (representing spatial average values for the whole ice tongue). In contrast, the radar-based measurements provide a spatial distribution with enhanced basal melt rates close to the grounding line - representing the zone of highest melt. Both the area-averaged estimates and point estimates can be reconciled with satellite-
based estimates of the basal melt rate by Wilson et al. (2017). Their bulk estimate gives values of 7.4 m $a^{-1}$ with maximum melt rates exceeding 50 m $a^{-1}$ near the grounding line. It is likely that the Wilson approach underestimates the bulk rate because the underlying method does not resolve the extreme melt rates near the grounding zone revealed by Zeising et al. (2023). Our modeling-based estimate shows a remarkable agreement to the mooring-based results. In their basic configurations, with ocean temperatures close to the observed ones, the area-averaged annual mean melt rates are 12.3, 9.2, 10.1 and 10.3 m $a^{-1}$
for the 2D fjord, 2D plume, 3D fjord and 3D ocean-sea ice model simulations, respectively. The 2D fjord model was used to explore melt rate sensitivities, displaying a linear sensitivity of 0.64 m $a^{-1}$ per 0.1° C to ocean temperature change. The tracer observation-based estimate of the melt rate of 8.6 m $a^{-1}$ was conducted during a time of particularly low AIW temperatures in summer 2016, that according to Schaffer et al. (2020) were about 0.2 – 0.3° C lower than the annual mean AIW temperature (August 2016-September 2017), which was used to derive the mooring-based estimate of 10.4 m $a^{-1}$. Therefore, applying the
simulated sensitivity of 0.64 m $a^{-1}$ per 0.1° C to the former estimate, an offset of 1.3 – 1.9 m $a^{-1}$ would be applicable to account for the temperature difference, bringing both estimates in even closer agreement with each other.

Estimates of subglacial discharge (given as a fraction of the sum of basal melt and discharge; see Table 1) have been provided from the mooring-based approach, yielding 11%. This is in reasonable agreement with results from the 3D fjord and 3D ocean-sea ice models (11% and 20%, respectively), where the absolute value of discharge is prescribed (using a constant
or time-variable flux). In the future, a seamless approach of linking the surface mass balance to the subglacial discharge via hydrological modeling would be desirable in order to use this as a temporally and spatially varying boundary condition for simulations of the ocean circulation and basal melt rate in the cavity of the 79NG.



In terms of ocean and subglacial fluxes inferred from the mooring-based observations, we find a transport of 270 mSv of AIW flowing toward the inner continental shelf, of which 46 mSv enter the cavity, resulting in a meltwater flux of 0.56 mSv

(which is 2 orders of magnitudes smaller than the AIW inflow volume transport), supported by a subglacial discharge of 0.07 mSv. On the continental shelf, we find that the meltwater flowing out of the cavity of the 79NG mostly stays within a depth range of 100–200 m. This is consistent with the meltwater plume detaching from the ice base before leaving the cavity - thus detrainment of waters from the plume taking place – something which the idealized 2D fjord model actually shows (Reinert et al., 2023). Once leaving the cavity, the meltwater quickly dilutes to values below the detection limit on the outer shelf,

meaning that the subglacial meltwater impact on the ocean stratification is limited to the near-coastal areas. Finally, looking on an East-Greenland-wide scale, the simulated near-shore sensitivity of the sea surface salinity to surface meltwater runoff (treating all meltwater as surface runoff) exceeds $0.5 \text{ g kg}^{-1}$ (compared to a case where runoff is switched off), with again the signal quickly dissolving away from the coast the shelf edge. Thus we expect the anomaly to drive an anomalous, narrow along-coast circulation, which may impact exchange of water between the fjords open ocean. On the west coast of Greenland, the

surface freshwater anomaly extends way offshore (e.g. into Baffin Bay) and is associated with a surface warming. Interestingly, (Oltmanns et al., 2020) found the strengthening of the ocean stratification associated with an increased freshwater runoff from Greenland to cause the ocean surface temperature to rise above freezing earlier within the season, affecting air-sea heat fluxes and thus mid-latitude weather patterns significantly. In the future, a more realistic treatment of meltwater runoff in model simulations would be desirable all around Greenland, in the sense that firstly it should be injected into the fjords at the correct

depths levels (glacier base rather than surface) and secondly the meltwater signature of icebergs melting offshore should be taken into account (Moon et al., 2018). We assume that this treatment, by spreading the meltwater signature both in the vertical and horizontal domains, might largely change the simulated ocean circulation response to meltwater runoff and ice discharge. As the runoff is strongly impacted by the prevailing ocean circulation, a more realistic approach could possibly even feed back onto ocean-driven glacier melt (Davison et al., 2020).

To conclude, the different in situ and remote sensing observations and model simulations together reveal a consistent picture of a coupled atmosphere-ice sheet-ocean system, that responds to the changing climatic conditions and that has clearly evolved in the past decades in all of its compartments.

Table 1: **Synthesis table containing the GROCE estimates concerning the mass balance of the NEGIS-79N-ocean system.** Numbers in the first row refer to the entries in Fig. 19. *Using the 79NG tongue base area of 1,700 $\text{km}^2$, the basal melt rates (m $\text{a}^{-1}$) in lines 14 - 20 can be converted into melt water volume or mass fluxes.

| No | Quantity | Value / Observation | Method |
|---|---|---|---|
| 1 | Annual mean surface air temperature; | $0.08°$ C $\text{a}^{-1}$ rise in surface air temperature (1979-2017); | i) Reanalysis data; |
| | Winter Warm air events (T>10° C) | 24% of events occur in January | ii) Regional modeling using WRF |





| No | Quantity | Value / Observation | Method |
|----|----------|---------------------|--------|
| 2 | Climatic mass balance of 79NG | Average is -0.74m a$^{-1}$ (2014-2018); annual range: -0.12 to -1.05 m a$^{-1}$ | COSIPY/NEGIS_WRF |
| 3 | Multi-annual average of maximum (peak) daily lake volume | $7.97 \pm 1.96 \, 10^8$ m$^3$ (2016-2022) | Supraglacial lake areas and depths derived from Sentinel-2 imagery |
| 4 | Tidal forcing | significant up to a distance of 10 km upstream from the grounding line | Viscoelastic simulations using COMice-ve driven by effective pressure from subglacial hydrology model CUAS (driven by observed tides) |
| 5 | Tidal forcing | makes up 6-30% of the strain in our simulation experiment | dito |
| 6 | Calving front retreat | ZI: retreat of 7 km (03/2013-10/2021); 79NG front shows no change | Landsat imagery |
| 7 | Simulated ice export of 79NG (across grounding line); acceleration in response to ice tongue geometry | 79NG: 11.9 Gt a$^{-1}$ ; 79NG enters mode of acceleration for tongue length reduced to <40 km | ISSM simulation |
| 8 | Mass loss rate of drainage basins of 79NG and ZI | ZI: -3.7 $\pm$ 1.3 Gt a$^{-1}$ over an area of approx. 90,000 km$^2$ (2010-2017); 79NG: -0.9 $\pm$ 0.4 Gt a$^{-1}$ over an area of approx. 110,000 km$^2$ (2010-2017) | Remote sensing: Combined gravity (GRACE) and altimetry |
| 9 | Solid Earth response to present-day ice mass change (vertical elastic displacement) | at GNSS sites displacement rates range from 2.4 x $10^{-3}$ to 7.0 x $10^{-3}$ m a$^{-1}$; maximum rate: 13.8 x $10^{-3}$ m a$^{-1}$ (at ZI calving front) | Modeling supported by GNSS measurement |
| 10 | Average annual freshwater runoff of FIIC | 11.40, 9.57, and 9.93 Gt a$^{-1}$ (2014-2017) | OGCM model forced with CRU, ERA5, and NEGIS_WRF atmospheric data, respectively |
| 11 | Mass loss of peripheral glaciers | FIIC: 2.8$\pm$0.7 Gt a$^{-1}$; NE Greenland peripheral glaciers: 8.1$\pm$4.3 Gt a$^{-1}$ (2010-2019) | OGGM Modeling |





| No | Quantity | Value / Observation | Method |
|---|---|---|---|
| 12 | NE Greenland projected ice volume of peripheral glaciers | 78±4% (SSP126) and 61±9% (SSP585); value in 2100 relative to 2019 (=100%) | OGGM Modeling |
| 13 | NE Greenland projected freshwater runoff of peripheral glaciers | 3922±315 Gt (SSP126) and 5576±826 Gt (SSP585) (2020-2100) | OGGM Modeling |
| 14 | Basal melt rate of 79NG (BMR/(BMR+discharge)) | 10.4±3.1 m a$^{-1}$ (89%) (summer 2016-summer 2017) | Mooring observations of annual mean ocean heat flux |
| 15 | Basal melt rate of 79NG (BMR/(BMR+discharge)) | 8.6±1.4 m a$^{-1}$ (<65%) (summer 2016) | Tracer (Helium, Neon) observations |
| 16 | Basal melt rate of 79NG (BMR/(BMR+discharge)) | 12.3 m a$^{-1}$ (90%); 14.3 m a$^{-1}$ (84%); 9.1 m a$^{-1}$ (98%); 15.3 m a$^{-1}$ (92%); 18.7 m a$^{-1}$ (93%) | 2D fjord model: present day; Subglacial discharge x 2; Subglacial discharge x 0.1; ocean temp. +0.5° C; ocean temp. +1.0° C |
| 17 | Basal melt rate of 79NG (BMR/(BMR+discharge)) | 10.1 m a$^{-1}$ (89%) | 3D fjord model: present day |
| 18 | Basal melt rate of 79NG (BMR/(BMR+discharge)) | 9.2 m a$^{-1}$; 8.1 m a$^{-1}$ | 2D plume model: Smooth ice base topography; rough ice base topography |
| 19 | Basal melt rate of 79NG (BMR/(BMR+discharge)) | 10.3 m a$^{-1}$ (80%) (1970 - 2021); Quadratic relationship of basal melt $M$ with ocean temperature $T$: $M(T) = 2.7T^2 - 4.3T + 10.7$; Square-root relationship of basal melt $M$ with subglacial discharge $D$: $M(D) = 1.2\sqrt{D} + 9.6$ | 3D ocean-sea ice model: FESOM2.1 |
| 20 | Basal melt rate of 79NG (BMR/(BMR+discharge)) | 7.4 m a$^{-1}$ (2011-2015) | Satellite measurement (Wilson et al., 2017) |
| 21 | AIW transport into cavity | 46±11 mSv (summer 2016-summer 2017) | Mooring observation |
| 22 | AIW water transport toward inner shelf at Ile-de-France | 270 mSv (summer 2014-summer 2016) | Mooring observations |



| No | Quantity | Value / Observation | Method |
|---|---|---|---|
| 23 | Meltwater fractions at 79NG calving front | Up to 1.8% (summer 2016) | Tracer (Helium, Neon) observations |
| 24 | Ocean surface salinity change off East Greenland | near-coastal decrease exceeding 0.5 when including Greenland freshwater (compared to reference run) | 3D ocean-sea ice model: FESOM 1.4 |

*Code availability.* The codes used to create the figures and calculations can be obtained from the authors upon request. The code of the idealized 2D fjord model for 79NG can be downloaded from Klingbeil (2023) and Reinert (2023).

*Data availability.* The data sources of most of the data sets forming the basis of this paper are given in the references provided throughout the text. Previously unpublished data sets used in this study include lake volumes (available upon request from KL), mass balance terms for the peripheral glaciers (available upon request from MS). The mooring data recovered in 2021 are available via https://doi.org/10.5281/zenodo.10469656 With detailed information available in the expedition report of HDMS TRITON (https://epic.awi.de/id/eprint/57629/, DOI: 10.57738/BzPM_0769_2023).

*Author contributions.* All authors contributed to the writing and data analysis. TK, TM, MS and AH coordinated the writing process.

*Competing interests.* One of the authors is a member of the editorial board of TC (Thomas Mölg). All other authors declare that they have no conflict of interest.

*Acknowledgements.* The authors would like to acknowledge the support provided by the German Federal Ministry for Education and Research (BMBF) within the projects GROCE (grant 03F0778) from 2016 – 2019 and GROCE 2 (grant 03F0855) from 2020 – 2023. This
funding provided the decisive basis to accomplish this study. The research work was also supported by the Deutsche Forschungsgemeinschaft (DFG), Special Priority Program (SPP) 1889 "Regional Sea Level Change and Society" (SeaLevel), as well as by the Helmholtz Society (Impulse and Network Fund). The authors gratefully acknowledge the computing time granted by the Resource Allocation Board and provided on the supercomputer Lise at NHR@ZIB as part of the NHR infrastructure. We thank the masters and crews of R/V Polarstern (Alfred-Wegener-Institut, 2017), the helicopter crews and the German weather forecasters for their great support of our measurements at
79NG and ZI during expeditions PS100 (2016) and PS109 (2017). Ship time was provided under grants AWI_PS100_01, AWI_PS109_03 and AWI_PS114_01. We thank the master and entire crew of HDMS TRITON and the Danish Arctic Command for their support to recover the moorings at 79NG in 2021.





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
