# Peer review of "The atmosphere-land/ice-ocean system in the region near the 79N Glacier in Northeast Greenland: Synthesis and key findings from GROCE"

_EGUsphere, 2024_

## Author Response (AR1)

Replies to reviewers

The authors would like to take the opportunity to thank the reviewers for their insightful comments that have helped to improve the manuscript. In the following we respond to each of the suggestions.

In addition to the reviewer suggestions, a number of other updates were done.
Firstly, there is the inclusion of the McPherson et al. (2024) study published a few of weeks ago that attributed a period of abrupt cooling of Atlantic Intermediate Water (AIW) observed at the 79N Glacier calving front between 2017 to 2021 to blocking-related cold air anomalies. This strengthening of the cold air export from the Arctic enhanced the ocean heat-loss of Atlantic Water (AW) water upstream in Fram Strait and the Nordic Seas and slowed down the circulation, intensifying the AW cooling. The ocean cooling was also connected to a reduction in the basal melt rates at the grounding line, suggesting far-field controls of local ocean-driven melt. As a result, we replaced the former single-panel Fig. 14 by a multi-panel figure including contents shown formally in Fig 13 (thus there is one Fig. less now).

Second, the results of the 2D plume model slightly changed. New simulations that account for friction in the calculations have been conducted. This led to a different response of basal melt rate to the smoothness of the ice base. Consequently, Figure 14 e) and f), the integrated values of basal melt in Table 1 and in Figure 18, and the values in section 4.11 have been modified.

Thidly, we added one co-author, Maria Kappelsberger of TU Dresden.

**Review 1**
Review of the manuscript "The atmosphere-land/ice-ocean system in the region near the 79N Glacier in Northeast Greenland: Synthesis and key findings from GROCE" by Kanzow et al.
This is an interesting and important paper that provides an overview of the results obtained through the GROCE project. GROCE focuses on in-situ observations, remote sensing data, and numerical modeling of the 79N glacier. As this is an overview paper, many details regarding the methods and results are provided in more technical papers published by the GROCE team over the past few years. The study area is very remote, difficult to access, and challenging due to weather conditions. Therefore, any field observations that can improve our understanding of the mass balance of the area and the dynamics of the 79N glacier are highly appreciated. In general, I think the manuscript is well-structured and provides a very comprehensive overview of GROCE. I recommend publication after minor revisions.
We thank the reviewer for this favorable assessment.

In general, the manuscript is well organized. However, try to avoid formulating questions in the text, such as those on lines 29 and 55, as it makes the manuscript look like a thesis.

The questions have been converted into statements.

Line 3:
"Even the glaciers draining the Northeast Greenland ice stream have been observed to retreat and thin."
replace with something like,
"In line with the rest of the Greenland ice sheet, the glaciers draining the Northeast Greenland ice stream have been observed to retreat and thin."

Done.
Line 8: "Our study is based on observations…" change to "Our study is based on field observations…"
Done.

Line 36-40: I find the estimate of Greenland peripheral glacier very high. 20% and 30 % is very high and not consistent with e.g. Khan et al (2022) who estimated 11 %. My guess is the different literature uses different definitions of peripheral glacier/ice sheet area.
Agreed. Change applied: "…Helm et al., 2014). The contribution of peripheral glaciers (i.e., glaciers dynamically disconnected from the inland ice sheet) to the total ice-mass loss of Greenland is significant, with estimates ranging from 11% to 20% (Bolch et al., 2013; Khan et al., 2022; Bollen et al., 2023; Kappelsberger et al., 2021), despite the fact that they cover only about 5% of the total ice area. This underlines the importance…"

Line 45: "…basal and frontal melt." Add reference.
Done

Line 48: delete "for a review" ??.
Done.

Line 58-59: "A temporary acceleration of ice mass loss in 2012 was.."
How about the extraordinary huge ice loss in 2019? (see e.g. Khan et 2022, Sasgen at al., 2019)
Done. A temporary acceleration of ice mass loss in 2012 was linked to atmosphere-induced surface melt in southwest Greenland (Bevis2019), followed by another acceleration in 2019 (Sasgen2019, Khan2022b), …

Line 71: along with (Mouginot et al., 2015) add a reference to Khan et 2014.
Done.

Line 71: "…largest remaining floating ice tongue…" replace with "…one of the largest remaining floating ice tongue…". I think Petermann Glacer has a 80 km long floating tongue too.
Both tongues have roughly the same length, but 79NG is wider. This has been clarified.
Change applied: Currently, the largest remaining floating ice tongue by area,...

Figure 3: color of GNET sites and peripheral glaciers is same. I cannot see the gnet site NORD (station north).
No change applied. All GNET sites (blue rectangles) are situated in ice-free areas. Thus, the GNET site at station Nord is clearly visible. Besides, the GNET sites are just added for completeness; we did not perform own analyses based on these data.

Figure 4: you have added name of TU GNSS station LAP1. Why not also add names of GNET station and AWS stations?
The intention here is to concentrate on the studies that have been accomplished in the framework of the GROCE project. Therefore, exclusively the station LAP1 is named in the map, since it was established during short field campaigns in 2021 and 2022. There are a number of further GNSS stations both by TU Dresden and GNET, which are explained in the legend (red triangles and blue rectangles, respectively); likewise for the AWS sites which are maintained by colleagues from The Netherlands.

Line 192: why not use a more updated version, e.g Simonsen 2021.

No change applied. The study performed by Kappelsberger et al. (2021) 3ocused on the time period 2010 to 2017 to achieve the best time overlap between the GNSS measurements, satellite altimetry and gravimetry. For this time period, also the results of an analysis of satellite altimetry were used for the Greenland Ice Sheet including Flade Isblink inferred by an own processing in cooperation with V. Helm (AWI Bremerhaven). Only for the peripheral glaciers (except Flade Isblink) the height changes were taken from the ESA GrIS CCI product where the methodology was described by Simonsen et al. (2017).  The products published by Simonsen et al. (2021) were not available at the time of publication of the Kappelsberger et al. (2021) study.

Line 201: I only see 7 TU GNSS sites on bedrock in fig 3? The text says 10 sites?
In Fig. 3 the TUD campaign sites on bedrock are marked by red triangles, from north to south: #1 at the southwestern margin of Flade Isblink, #2 to 5 in Kronprins Christian Land (north of 79NG), #6 and 7 at the coast north of 79NG (Holm Land and northern coast of 79NG fjord entry), #8 western Lambert Land (there the rectangle of the newly established permanent site overlaps, since it is very close to the campaign site), #9 at an island southeast of Lambert Land (Franske Oer), #10 at an inland nunatak more than 100 km south of Lambert Land.

Line 395: "We now move in…" delete 'in'
Done.

Line 431: "drainage basins of ZI and 79NG, respectively"
Please show drainage basin of  ZI and 79NG in figure 10a.
Figure updated. Drainage basins are now shown in the respective subfigures (showing NE Greenland at larger scale).

Line 433: replace "ist" with "is"
Done.

Figure 14 caption: "Note that the ApRES drifted with the ice away from the grounding line over time and was relocated to the original starting point in July 2018."
State in the caption how much the ApRES drifted.
This sentence is not part of the caption anymore. In the method section, we state that the ApRES has moved 2.5 km before it was relocated.

Line 720: Data availability.
Since this is an overview paper, I strongly suggest you provide information (link to https) exactly where data presented in paper can be found. I know most of the data is available through other GROCE publications, however, it will be nice to have all links gathered is this study.
We agree. We have added an appendix to the manuscript detailing the references of the GROCE data sets both shown in the figures of the manuscripts and their contribution to the estimates in Table 1.

**Review 2**

Review of Kanzow et al. 2024: "The atmosphere-land/ice-ocean system in the region near the 79N Glacier in Northeast Greenland: Synthesis and key findings from GROCE" (egusphere-2024-757)

This paper synthesizes the findings of the GROCE project, which used observations, models, and remote sensing approaches to investigate virtually all aspects of the coupled ice-earth-ocean-atmosphere system in the vicinity of 79 North Glacier in northeast Greenland. This project has been a huge undertaking, producing a number of publications, and putting those papers in conversation with

one another and summarizing the major findings is a significant contribution. As those papers have already been published, I have only minor comments that I hope will help make this manuscript more readable. Once the authors have had the chance to address these, I look forward to seeing this work published.

We would like to thank the reviewer for the encouraging statements.

I noticed that at least one of the GROCE paper citations (Zeising et al. 2023) directed to the pre-print — it might be good to go through these recent publications and make sure the DOI directs to the final work.

Done.

Line 82: Please define GROCE (I think it's currently defined only in the abstract).

Done.

Section 2 (line 94-102): At the moment the information given in this section seems like it could fit into Section 1 or 3. A few more details about GROCE could be added here, e.g. the years that GROCE was operating with an overview of the major field campaigns (leading into the details given in Section 3). More details could also be given about the geometry of 79NG, e.g. the grounding line depth, max. fjord depth/width, and ice tongue thickness (building on the overview given in Section 1 and the schematic in Figure 2).

We re-arranged (and partly re-wrote) the text between the end of Section 1 and Section 2. The latter now consists of four paragraphs and has more weight than in the initial manuscript (also responding to comments of reviewer 3). This includes information on the expeditions as well as on the geometry of the 79NG. We also took this occasion to formulate and articulate our aims with this article more clearly, which is now at the end of Section 2.

Line 98: Remove (ii) (or add corresponding (i)).

Done.

Line 162-163: I suggest rewording as "The feature importance assessment shows…"

Done.

Line 164: I would suggest moving the reference to Fig. 4 to end of following line, because Fig. 4 shows two examples of frontal positions (but not a comparison of single-band to multispectral input data).

Done.

Line 165: Which years does this analysis cover?

The following change was applied. "As a result, between 2013 and 2021 we could infer …"

Line 166: It's unclear to me what is meant by "in a better way" — maybe rephrase as "in order to quantify the change in frontal position, a box method…"

Done.

Line 171: Change "subsequent" to "sequential"

Done.

Line 188: Define "GrIS" or type out acronym as it is not used elsewhere (except Line 434).

Done.

Line 189: Might be helpful to point out the location of FIIC in Fig. 3 as this is its first mention in the text.

Done.

Line 210: Remove extra "(NE)".
Done.
Line 253-256: Here it is stated that the instrument was initially deployed in 2018, then relocated in 2020, and recovered in 2022, but in Fig. 14 the time series extends from 2016-2020 (relocated in 2018) — are these referring to the same instrument? Please clarify.
Thank you for noticing. We made a mistake here and described a different ApRES than the one shown in Fig. 14 (now Fig. 13). We have corrected this.

Line 257: What is meant by "hinge zone"?
The hinge zone is the zone downstream the grounding line where the ice is bent by the tides. Since the use of the term is not necessary here, we have simplified the sentence: "The instrument was deployed about 6 km downstream from the grounding line."

Line 271 & 278: Change "Instead…" to "In contrast…"
Done

Line 337: Change "Atlantic Intermediate" to "AIW"
Done.

Line 359: Clarify that this is referring to intraseasonal (not interannual) variability?
Text changed and clarified

Line 360-364: Consider including a reference to Figure 6 here.
Good point. We include a reference to Figure 6 now and also revised the text to make clearer that Turton et al. (2021) looked at melt pond areas (versus melt pond volume that is shown in Figure 6).

Line 380: This sentence seems to be missing a word/words — "one manifestation of _____ at the atmosphere/glacier interface"
Thank you, we see the confusion and changed the text.

Line 386-387: I found the wording around "maximum (peak) daily lake volume" a bit confusing here…it makes sense to me that in the caption of Figure 7, the timeseries are accurately described as "daily lake volumes," but is it necessary here to specify that these are based on daily values? As a non-hydrologist, my initial interpretation of this phrase was that it was the average of all daily maximum values during every day of the melt season across all years, as opposed to the average peak volume across all years. I would consider omitting the word "daily" in line 386, and replacing "maximum daily" with "peak" in line 387 and the caption of Figure 6.
We followed this suggestion and changed the text accordingly.

Line 389-390: Again I'm confused by the phrase "total daily lake volume" — to me this suggests a cumulative or average volume, but the value in the text ("around 8.6*10^8 m^3" in both 2020 and 2021) appears to be the peak lake volume in 2021. (The peak lake volume in 2020 is 9.9*10^8 m^3.) If this is meant to compare peak volumes please say this explicitly (or if it is something else please clarify).
Correction applied.

Line 395: Change 79N to 79NG
Done.

Figure 8: Consider adding a scale bar or axis labels for satellite images. Clarify that y-axes in time series are anomalies in axis label or caption.

There is a scale bar in the maps, and we clarified in the caption that the y-axis shows the area change with different scales.

Line 432: Change "ist" to "is"
Done.

Line 433: What accounts for the residual 4%?
There was a mistake, it should be 87% for the Greenland Ice Sheet (and 13% for the peripheral glaciers). Number corrected in the manuscript.

Line 434: Define "GrIS" or type out acronym as it is not used elsewhere (except Line 188).
Done.

Line 475: Clarify in text that SSP126/585 are low-/high-emissions scenarios as these have not been defined previously.
Done.

Line 518-521: At what depth/distance along-fjord does the plume reach neutral buoyancy and detach from the ice tongue — is this significantly deeper than/upstream of the calving front? Or are the mid-depths where the exported meltwaters are observed (e.g. in Section 4.13, line 608-10) consistent with the keel depth of the ice tongue near the front?
The upper depth limit of the exported waters is roughly consistent with the keel depth as follows. In the 2D fjord model, the plume has a height between 3 and 5 m and detaches from the ice tongue around 42 km downstream from the grounding line, where the ice draft is 95 m (exceeding the ice draft at the calving front by just 20 m). In the 3D ice-ocean model, deflection by the Coriolis force leads to a just 3 km wide plume detached from the ice base, residing between 100 and 280 m depth at the southern margin of the cavity. The plume significantly widens horizontally near the calving front. This information has been added, distributed in several sentences.

Line 534-535: Consider referring to the map in Figure 4 that shows the location of these measurements.
Done.

Line 538: Why does the difference between the median and 95% quantile reflect spatial variability? Is this related to the roughness?
The melt rate analysis is based on nadir and off-nadir melt rates from side reflections. If melt rates are nearly homogeneous near the measurement site, the difference between the median and the 95% quantile is small. A large difference indicates that both low and high melt rates occur near the measurement site. This is primarily due to changes in basal topography. In the revised version, we split the sentence into two: "Fig. 13c presents a timeseries of basal melt rates inferred from the ApRES measurements and reveals strong temporal variability, with no clear seasonal cycle. The difference between the median and 95% quantile also reflects the spatial variability as a large difference indicates the presence of low and high melt rates near the measurement site at the same time due changes in basal topography."

Line 542: The decline in AIW temperature is not shown explicitly in Fig. 14, is this meant to refer to the deepening 1.2° isotherm in the previous figure?
That is correct. We referred to Fig. 14 where it should have been Fig. 13. The connection to the isotherm is made clearer now. Fig. 13 was exchanged by the one published by McPherson et al., (2024). We now highlight the 1.0°C isotherm (rather than the 1.2°C one), which make no difference

contents-wise. With the inclusion of the McPherson et al. (2024) study, we can also make the link between AIW temperature and melt rate response much clearer. The following change was applied. "The reduction coincided with the observed decline of the AIW temperature near the calving front, manifesting itself in the deepening of the 1°C isotherm (Fig. 13 a,b). McPherson et al. (2024) found that changes in the temperature of the inflowing AIW at the calving front are expected to drive changes in the melt rate at the grounding line a few months later (see also Sect. 4.12)."

Line 546-547: Add reference to Fig. 15d.
Done.

Line 549-550: Add reference to Fig. 15a.
Done.

Line 565: Figure 10 does not show changes in ocean temperature so it's unclear why it is referenced here.
Reference corrected.

Line 582: Change to "the subsurface AIW transport"
Done.

Line 591: Missing a word? This should maybe read "also shows that not only the AIW temperature…"
Indeed. Corrected.

Line 713-714: I'm not sure that it's true that the *runoff* is affected by the ocean circulation; the Davison paper cited here is rather about submarine melt enhancing fjord circulation and thus heat transport towards the glacier.
Indeed, that is what we intended to say. The statement now reads: " … change the simulated ocean circulation response to meltwater  and ice discharge. As the submarine melt rate is strongly impacted by the prevailing ocean circulation, a more realistic approach of meltwater injection could possibly even feed back onto ocean-driven glacier melt (Davison et al., 2020)."

Line 717: Change "compartments" to "components"?
Done.

Table 1, number 5: Method should say "ditto" (or write out as "same as above/no. 4")?
Done.

**Review 3**
General Comments / Summary
This paper reviews the GROCE project, a multidisciplinary research study on the 79N Glacier, Zachariae Isstrom, and the peripheral ice in the NE Greenland region. The GROCE project included observations (both in situ and remotely-sensed) and simulations of the ice, ocean, land, and atmosphere. Overall, the paper summarizes the research published to date from GROCE, most of which have come out in process studies. Thus, the goal of this paper was to provide a synthesis of these results and put the whole system in context.
My main takeaways from the paper were that 79NG and ZI differ significantly and GROCE nicely sampled these and produced useful comparisons for why those systems might differ. Second, the peripheral glaciers surrounding 79NG and ZI cannot be ignored, at least in terms of SLR potential. There are many, many other details in the paper that are interesting nuggets. However, it is not clear at all whether these are 'new' tidbits being discussed here or whether they were originally presented in the earlier papers.

We think that a remarkable outcome of the study is that the different in situ and remote sensing observations and model simulations together reveal a consistent picture of a coupled atmosphere-ice sheet-ocean system, that responds to the changing climatic conditions and that has clearly evolved in the past decades in all of its components. This becomes only visible when piecing the different parts together. We also find it remarkable, that the impact of the far-field ocean forcing affects the melt rates of the 79NG much more strongly than forcing on the continental shelf or subglacial runoff.

My main critique of the paper is that it is unnecessarily long and that the main takeaways (see above) are hidden at the back of the study, i.e., you have to get through 18 Figures and a lot of methods before finally getting to the synthesis. I would love to see the structure of the paper flipped somehow and relying more on the published papers, since it's not clear at all whether they are adding anything to those. See one of my specific comments below for more details. I do not mean to detract at all from the impact GROCE has had and the observations/modeling they have done are critical to our improving our understanding of the ice sheet. I just do not see the benefit of the paper as written, other than as a repository for their project.

We thank the reviewer for the thoughtful, thorough and demanding review. We highly appreciate the intention, to put a larger emphasis on discussing the results of GROCE in the context of Greenland and Antarctic literature. We appreciate the suggestion to distinguish previously published from new findings more strongly. Both points were taken up during the revision process.

At the same time we also ask for understanding to not change the basic approach of listing individual GROCE study results. In fact, the original version of the manuscript did rely to the largest part on published results. In our opinion, this is also a service of a synthesis article for readers (and more than a repository), so they do not have to go into 20+ papers to learn about all these results.

We would further argue that the reader does not need to wait to the very end of the paper before the integration / synthesis starts. A lot of integration work has been accomplished in figures 3-5 (methods; Chapter 3), bringing visually together all the observational sites and areas as well as the scales covered by the different models. In terms of the Figures 6-18 contained in Chapter 4, Figures 6, 7, 8, 9, parts of 12, part of 14, and 17 have been specifically composed for this manuscript. Some of them are based on published analyses and data sets, yet they offer new perspectives, allowing the different publications to communicate with each other. We point this out more clearly, now.

To accommodate the main critique, we have taken several steps during the revision.
Firstly, we have defined our goals in chapter 2 more clearly. They now read: ". The motivation of this article is to bring together the many individual, in-depth studies of GROCE, based on both published research and new data. From this we derive the main aim, which is to provide (i) a process-based, holistic understanding of the ice mass balance of the NEGIS-79NG system and (ii) a quantitative assessment of the involved mass balance components. Our approach incorporates a synthesis of processes from atmosphere to ice, to ocean and to lithosphere."

Secondly, we agree, that it was not obvious, which methods / results were published and what was an update to the published work. In the revised MS, we skipped the term "new" throughout. Also, each Figure caption in chapter 4 now contains a statement regarding the "novelty", meaning whether it shows published data, whether the data set was extended, or whether the data set is new.

Thirdly, we have shortened the methods section by 30 lines (12%) despite reviewers' requests for more information.

Fourthly, we added links to Greenland-wide / Antarctic perspectives by placing out results in the context of external studies.

Lastly, we added an appendix detailing the reference to the data sets underlying the presented figures and GROCE publications.

Specific Comments
1) As mentioned above, it was not clear to me what methods presented in Section 3 were new. Many of the subsections (e.g., 3.1, 3.3., 3.4, 3.6, 3.7, 3.9, etc.) are titled 'new' or 'refined' approaches to their respective measurements. However, upon reading those sections, it seems like a recapitulation of previous papers without any clear indication of what is new. Thus, the reader is left with having to go back to all the previous papers anyway, significantly lessening the benefit of this review paper.
We have removed all occurrences of "new" and "refined" in the subtitles of chapter 3.
We have shortened the method by 12% (despite adding information demanded by the reviewers).
In the figure captions of chapter 4, we now state where data sets have been added compared to published results.

In my opinion, the paper would benefit from clearly stating up front what synthesis is new, i.e., what are they trying to do in this paper that differs from all the previous work (besides just summarizing everything). As it reads now, there is a lot of self-citation outside of the introduction- it would be helpful/useful to see the authors compare ZI and 79NG not just to each other, as they do, but also to other process studies and systems around Greenland (and Antarctica, as commented below). What can we learn from GROCE to help us observe/understand other systems? It's such a large, multi-faceted project that I cannot envision doing it everywhere, so what are the basic takeaways or observations we need to do this elsewhere?
We have now embedded our results into the pan-Greenland and Antarctic literature, following specific comments of the reviewer (see below). For instance, for Antarctica, Pine Island Glacier seems a suitable parallel: small, high melt rates due to warm water in cavity, large throughput of ice, drains an over-proportionally large area of ice sheet. We make this point now in chapter 5.

2) When reconciling the observations and model estimates of basal melt, one must always be careful of how the models are calculating melt rates. I did not see any discussion of how the models they are using parameterize melt rates. Are they using standard parameter values or are they tuned to fit the observations? This would help make the comparisons more useful/novel later on.
The melt rate formulation in the two ocean models described here is based on the three-equation system. An important parameter in the basal melt formulation is the drag coefficient $C_D$, which is not well constrained and sometimes used to tune the models. In Wekerle et al. (2024), we used a rather low value representative for "warm" ice shelves that has also been used in other Greenland ice shelf modeling studies. We did not specifically tune the model to fit the observations. A sensitivity study, however, revealed that the magnitude of basal melt rates is strongly dependent on the parameter. We added the following text in the manuscript:
*"However, it has to be noted that the basal melt rates obtained by the ocean models are also subject to uncertainties. The simulated basal melt rates were computed based on the three-equation system. An important parameter in this formulation is the basal drag coefficient, which is not well constrained by observations and sometimes used to tune the models. Indeed, sensitivity experiments showed that the magnitude of the simulated basal melt rates of the 79NG strongly depends on the choice of the basal drag coefficient (e.g. Wekerle et al. 2024)."*

3) I was surprised to see so little reference to the Antarctic literature, since ice shelf/cavity circulation and basal melt processes are more common there. Is anything fundamentally different about the NE Greenland systems or can we transfer the knowledge from Antarctic systems there? Or vice versa, given the scope of GROCE.
Good point. We have added a discussion of the melt rates' sensitivity to subglacial discharge, comparing to Getz and Pine Island Ice Shelves in Antarctica. Except for differences in the absolute

numbers here, which can be attributed to the different volumes of runoff available, we tend to argue that ice-ocean interaction follows the same principles / relationships on both hemispheres.

4) Figures 1 and 2 seem like copies of figures produced elsewhere in the literature. Figure 1 I find more useful than Figure 2. Figure 2 as it stands does not help me see any of the interactions or feedbacks in processes that they discuss.
No change applied. It is true that similar figures exist in the literature, but these differ in important details. Some aspects of GROCE are specific, so that we decided to purposefully design our own figure version. Figure 2 includes many of the processes we specifically refer to. We made 11 references to it in the originally submitted manuscript. In this regard, Figure 2 is central for the understanding of the manuscript and the GROCE project as a whole.

5) Throughout the paper, the authors use the ref Straneo et al 2010 to point to 'warming' or AIW around Greenland. However, that reference is really just a process study of how Atlantic Waters circulate in a Greenland glacial fjord. There is nothing in there about 'warming' of the AW. There must be better references out there for this...
Agreed. We switched the Straneo reference to Straneo et al. (2013) which makes the explicit link between widespread ocean warming around Greenland and glacier retreat. We also added Wood et al. (2021) on this issue.

6) Section 2 as written is superfluous. It is short and everything in it is contained in the next section 3. I would either delete it or give more details on GROCE in this section.
We re-arranged (and partly re-wrote) the text between the end of Section 1 and Section 2. The latter now consists of four paragraphs and has more weight than in the initial manuscript (also partly responding to criticism of reviewer 2). We also took this occasion to formulate and articulate our aims with this article more clearly (see critique above), which is now at the end of Section 2.

7) In Section 3.5: 'a thorough assessment' of what? And why? What did the assessment show? You state how many flow velocity fields and frontal positions you could infer, but it's not clear if this is better than other methods? The final sentence in that section is awkwardly worded, so I would consider rewriting/simplifying it.
The assessment was carried out w.r.t. the achievable accuracy when including diverse data. The resulting feature importance was discussed in detail by Loebel et al. (2022). We revised the wording and added one sentence to clarify the statements made.
The accuracy assessment and comparison to other data products were dealt with in detail by Loebel et al. (2024) [which is the finally published paper following the discussion paper Loebel et al. (2023) – reference was also updated]. It can be stated that the quality of our automatically inferred frontal positions is comparable to that of manually delineated fronts. It is a major advantage that our method allows to achieve a considerably higher extractions rate than other (automation) methods, and, thus, to infer time series of frontal positions resolving long-term, seasonal and subseasonal variations. The paragraph has been revised accordingly (including reformulating the last sentence).

8) Section 3.6: the section title suggests sensitivity studies were done in a novel way here. True or done in the previous papers? I don't remember reading later on in the paper any discussion of sensitivities to the choice of bathymetry (Bedmachine 4 vs. 5), calving locations or magnitude, or any other model parameters.
The term 'novel' may be a bit misleading. In a previous paper (Humbert et al., 2023), sensitivity studies were conducted on the influence of different positions of the calving front on ice discharge. Here, we described the three positions of the calving front: (1) after the calving of an iceberg, (2) after retreating to a 45 km long ice tongue, and (3) a complete collapse of the floating tongue. These cases are discussed in Section 4.6 and shown in Fig. 9 (which is based on the model simulations of Humbert et

al. (2023) yet was specifically composed for this MS). Indeed there is no sensitivity test re Bedmachine 4 vs. 5.

9) By start of Section 4 (16 pages in!), I am still not sure what the point of this paper is. There are so many methods, processes, model simulations (2D, 3D, ocean, atmosphere, etc.), and foci, that I am lost. I think shortening the methods or rearranging the structure somehow of the paper will make it much more impactful.

We hope that more clearly formulating the goals of the study in chapter two removes some of the concerns. We have shortened the methods. Basically the methods have all been published (as is clearly stated in the introduction to chapter 3).

10) Section 4 keeps going as Section 3 did, which is a long list of results of each different method/study. What I really want from this review paper is a synthesis of the methods and a more thorough understanding of how 79NG/ZI work, how they compare to other systems in Greenland, and discussion of ways forward for observing/modeling these systems. Section 4.1, for example, seems like a rehash of the Turton et al papers. The last sentence in that section finally hints at something new, which is a comparison of the atmosphere to the lake area/volumes. This is interesting!

As stated above, we have made it clearer what is published and what is not. Basically the methods have all been published (as stated in the introduction to chapter 3). Over the course of chapter 4, based on the specific comments of the reviewer (#3, 16, 19, 20), we have embedded our results into the context of Greenland literature and (in chapter 5) also Antarctic literature.

11) Section 4.2 ends with a statement that "Fig. 6 summarizes the above-mentioned relations". What relations? Why not just reference Fig. 6 as you go?

We meant the "causal" relations discussed in the preceding sentences. However, we took the advice and now just reference Fig. 6 (and also changed the text slightly).

12) Section 4.3 seems new and interesting. Sections like this should be the focus of the paper. However, I don't see in Fig. 7 the evidence for the statement that 2020 and 2021 have similar total daily lake volumes? Is this cumulative or an average?

We agree that the description for Fig. 7 was not precise enough and revised the text (inserted date to which the comparison was meant to refer to).

13) Figure 8: would be useful to have the area curves on the same vertical axes for comparison.

We intentionally used different scales for the area (at the y-axis) since the area change of the 79NG would not be visible when plotted in the same scale as that of Zachariae Isstroem. The caption of Fig. 8 was revised accordingly.

14) Section 4.5: not sure why two glaciers near each other should have the same calving style. Maybe you mean retreat here? Calving style would seem to be dictated by size of the glacier (and whether it's an ice shelf, tidewater, or alpine glacier).

We agree that the statement at the beginning of this section is somehow imprecise. It is revised accordingly.

15) Section 4.7: These are all published results right? Or what is 'refined' about the mass balance estimates here? Same for Section 4.8. Please highlight what is new or make it clear this is already published and focus more on the synthesis of these different components of GROCE.

Of course, the reviewer is right, these findings are based on the results published by Kappelsberger et al. (2021). The term 'refined' was originally used to denote the inference of the ice-mass balance with a higher resolution in NE Greenland as well as the proper treatment of the peripheral glaciers

(especially of Flade Isblink). Heading of Sec. 4.7 was changed accordingly, replacing refined by 'high-resolution'.

16) In Section 4.9 and elsewhere for freshwater fluxes to the ocean, one useful comparison might be with the Bamber et al. datasets/products. How does this compare to what projects like O-SNAP suggest would be potentially significant FW flux to the subpolar N Atlantic in terms of modifying the AMOC?

This is a difficult task, which would deserve its own review article. The issue continues to receive a lot of attention, both scientific and societal, without really having been solved. We added the following statement to the end of chapter 4.9. We hope it does some justice to the topic.

"According to Bamber et al. (2018) the bulk (solid and liquid) freshwater flux from Greenland into the ocean has increased from the 1980s to the late 2010s by about 400 Gt a^-1 (corresponding roughly to 40 %). Given that the late 2010 rate is still just 20% of the ocean freshwater flux from the Arctic Ocean into the North Atlantic, the impact of the Greenland freshwater flux on the Atlantic thermohaline circulation has likely not yet been significant (Böning et al., 2016). Still freshening of the surface waters by Greenland freshwater has already been shown to episodically counteract the effect of wintertime cooling in terms of deepwater formation in the North Atlantic (Lozier, 2023), such that the Greenland impact might be emerging (Böning et al., 2016; Lozier, 2023). Martin et al. (2022) suggest that an increase in Greenland freshwater fluxes by roughly 1600 Gt a^-1 might reduce Atlantic thermohaline circulation by not more than 15%. Projected peak freshwater fluxes of Northeast Greenland peripheral glaciers of 50 – 80 Gt a^-1 occurring in the second half of this century may contribute to such a Greenland-wide impact, but are unlikely to be of leading importance."

We repeat the essence of this in one sentence in chapter 5.

17) Figure 13: The inflowing AIW is much deeper than the grounding line. Do you have ideas on how this warmer water gets mixed up? Is it the plume? Icebergs? **This is where it might be useful to discuss these ice shelf systems more in relation to Antarctic systems.**

The glacier is grounded at 600 m depth (Zeising et al., 2023), yet near the grounding line the ice rises steeply, and there are steep channels, too, depending on the exact location. Therefore, the ice draft shown in Fig. 13 should not be confused by the maximum grounding depth. Thus, the AIW inflow is shallower than the grounding depth, where a meltwater plume entrains the AIW. In the caption of Fig. 13 we added the sentence "Note that the maximum grounding depth of the 79NG slightly exceeds 600 m (Zeising et al., 2024), while the ice draft shown here corresponds to the local ice draft at the position of the instrument."

As such, we don't think we need Antarctic literature here. Apparently, the reviewer got a wrong impression about what the near-GL ice draft is, and with the above reply the issue should be clarified. We also used the term "Local ice draft" in caption of Fig. 13c now to avoid confusion.

18) Lines 515-520 in Sect 4.10 need some specifics. That is, what depths do the plumes equilibrate at? This would be helpful information later on when you discuss the spreading of meltwater around Greenland.

In the 2D fjord model the plume has a thickness between 3 and 5 m and detaches from the ice tongue around 42 km downstream from the grounding line where the ice draft is 95 m (exceeding the ice draft at the calving front by just 20 m). In the 3D ice-ocean model, deflection by the Coriolis force leads to a just 3 km wide plume detached from the ice base, residing between 100 and 280 m depth at the southern margin of the cavity. The plume significantly widens horizontally near the calving front. This information has been added, distributed in several sentences.

19) Sect. 4.11 has some interesting tidbits that seem novel: the fact that AIW feeds the basal melt rates and not the seasonal discharge. Does this make it different than other Greenland systems?

In our answer to comment 20 (below) we argue that the same kind of dependency of basal melt rates on AIW temperature and subglacial discharge we inferred for the 79NG has been found at other Greenland ice shelves as well (e.g. Ryder and Petermann Glaciers, Wiskandt et al. 2023, Cai et al. 2017). Ice shelves, or ice tongues, are exclusively present in northern Greenland. The relationships are slightly different for tidewater glaciers, mostly found in the southern, eastern and western parts of Greenland. See answer to comment 20 for the implementation.

Also, we added a sentence (as final sentence) to chapter 4.12 after "Wekerle et al. (2024) conclude that the temporal evolution of Atlantic Water in Fram Strait is the main driver of changes in basal melt rates of the 79NG over the past 50 years." with direct relevance to the pan-Greenlandic perspective requested here, as follows.
"This result may predominantly hold for Northeast und South Greenland outlet glaciers, while for the Northwest Greenland ones, the increase in subglacial discharge may have been the dominant factor for the increased submarine melt rates at the calving fronts (Slater and Straneo, 2022)."

20) Can you compare the Reinert et al 2023/2024 conclusions about melt rates and their dependencies to other papers/systems/theory out there? That would be a useful addition to the literature here.
Other studies of ice shelf basal melt came to similar conclusions regarding the relationship between basal melt and thermal forcing/subglacial discharge. For tidewater glaciers, however, a cubic relationship between basal melt and subglacial discharge was observed. We added the following text:
"Other studies of ice shelf basal melt came to similar conclusions regarding the relationship between basal melt and thermal forcing/subglacial discharge. An above-linear dependency of basal melt on thermal forcing and a square-root dependency on subglacial discharge has been revealed for Ryder Glacier and Petermann Glacier *(Wiskandt et al. 2023, Cai et al. 2017)* and for the Amundsen Sea ice shelves (Jenkins et al., 2018). For tidewater glaciers however, where the subglacial discharge can be much larger than the submarine ice melt, a cubic relationship with subglacial discharge *(Slater et al. 2016)* has been revealed."

The agreement of the dependencies among North Greenland floating ice tongue glaciers is also briefly mention now in chapter 5.

At the same time, we updated the results regarding the impact of the under-ice roughness of the melt rate. The qualitative results stay the same, but the values are slightly lower. This part now reads:
"An experiment featuring a smooth ice base geometry revealed a somewhat lower area-averaged melt rate of 4.3 m a^-1 compared to that resulting from the rough geometry 8.3 m a^-1 (compare Figs. 15 e,f). The latter features a clear imprint of the locations of the subglacial channels on the distribution of melt rates across the tongue (Fig. 15f)."

21) Fig. 19 and Table 1 contain a lot of useful numbers/information. However, I find Table 1 to be overly complicated. It could be organized better- maybe with subheadings on each topic, e.g., basal melt, AIW transports, etc.?
As suggested, we added subheadings in Table 1 to make it better readable.

22) In the abstract, I think the main issue with this paper is hinted at- it says, 'here we present a comprehensive study of processes' but then in the next sentence 'the focus is on 79N glacier'. I think this paper would benefit from doing one or the other and not trying to state everything that GROCE has found- as written, I found myself wanting (and needing) to go back to all the original studies for details and discussion. If I wasn't reviewing the paper, I would have skipped ahead to Fig 19 and Table 1.
No change applied. We think it is of particular interest not to view the 79N Glacier system in isolation, but also in the context of the neighboring ZI and the peripheral glaciers (eg. FIIC) in the area. This has

been a core aspect of the GROCE project. The common tie of the study is the mass balance. By stating the goals of the study more clearly in Section 2, this partly resolves the criticism, we hope. Also, having made now clearer, what is based on published results, should help the reader not having to go back to all published studies.

Technical Corrections
Line 26: more references here that are not self-citing would be useful, as there are many from OSNAP and other projects.
We added the recent OSNAP review paper of Lozier (2023), which seems a good choice, as it offers multiple references to very recent AMOC studies in the subpolar North Atlantic.

Line 37: 25% of what?
Here, we refer to the peripheral glaciers' contribution in comparison to the total ice-mass loss of Greenland. Statement was slightly revised to be correct.

Line 42: Again, maybe better references for warming of Greenland's waters? Or at least more recent, since Murray and Straneo are process studies about how AW gets to the fjords (and this might be very different than for ice shelf cavities).
As stated above, we switched the Straneo reference to Straneo et al. (2013) which makes the explicit link between widespread ocean warming around Greenland and glacier retreat. We also added Wood et al. (2021).

Line 64: No Antarctic references here? There is a ton of work done on ice shelf cavities.
Wei et al. (2020) claim to have demonstrated that subglacial discharge is aligned with basal channels and locally enhanced melt rates under Getz Ice Shelf. We come back to this study in the conclusion. For the more general statement we make here this finding is too subtle. The following study, however, is worth mentioning in the context discussed here. Nakayama et al. (2021) demonstrate that subglacial runoff becomes important [here defined as exceeding the 10% difference margin] for the total Pine Island Glacier melt rate only when it is enhanced by a factor of 10 and more compared to the best [model-based] guess. Nakayama (2021) actually explicitly states that the effect is much smaller than for Greenland. So here we added the sentence: "For Antarctic ice shelves the impact of subglacial discharge on basal melt rates appears to be less important (Nakayama et al., 2021)."

Line 159: 'Comprises' might be the wrong word here. Does it comprise the data or combine the data or use the data?
Agreed. Changed to "Combined".

Line 315: Earlier you used Bedmachine v4. Is there a difference in this region of Greenland?
There is indeed a difference between version 4 and 5. This is taken from the Bedmachine Documentation (https://nsidc.org/sites/default/files/documents/user-guide/idbmg4-v005-userguide.pdf): " *Version 5 added bathymetry measurements from the Alfred Wegener Institute, Helmholtz Centre for Polar and Marine Research, in front of the Northeast Ice Stream, ice thickness for ice caps from Millan et al. (2022), and bathymetry data from the 2021 OMG campaign. A geoTIFF file of bed elevation was also added. Version 5 was only released in September 2022, so studies conducted earlier (e.g. Humbert et al. 2023) used version 4* ".* An important consequence for us is that BedMachine v4 shows no ocean connection between Dijmphna Sund and the 79NG fjord (the connection is blocked by grounded ice and land). BedMachine v5 shows a water connection between the 79NG fjord and Dijmphna Sund (which we know to exist).
No change applied.

Line 582: delete 'the' in front of AIW.
Done.

Line 609: These depths were never specified earlier on in the discussion of the meltwater plumes...
The depths have been added in Sect. 4.10.

Line 664: Illustrated in... ?
 We deleted this (was a leftover that should have been deleted prior to submission).